# Decomposable Non-Smooth Convex Optimization with Nearly-Linear Gradient Oracle Complexity[*]

**Sally Dong**
Computer Science & Engineering
University of Washington
Seattle, WA 98195
sallyqd@uw.edu

**Haotian Jiang**
Computer Science & Engineering
University of Washington
Seattle, WA 98195
jhtdavid@uw.edu

**Yin Tat Lee**
Computer Science & Engineering
University of Washington
Seattle, WA 98195
yintat@uw.edu

**Swati Padmanabhan**
Electrical & Computer Engineering
University of Washington
Seattle, WA 98195
pswati@uw.edu

**Guanghao Ye**
Department of Mathematics
Massachusetts Institute of Technology
Cambridge, MA 02142
ghye@mit.edu

## Abstract

Many fundamental problems in machine learning can be formulated by the convex program

$$\min_{\theta \in \mathbb{R}^d} \sum_{i=1}^{n} f_i(\theta),$$

where each $f_i$ is a convex, Lipschitz function supported on a subset of $d_i$ coordinates of $\theta$. One common approach to this problem, exemplified by stochastic gradient descent, involves sampling one $f_i$ term at every iteration to make progress. This approach crucially relies on a notion of uniformity across the $f_i$'s, formally captured by their condition number. In this work, we give an algorithm that minimizes the above convex formulation to $\epsilon$-accuracy in $\widetilde{O}(\sum_{i=1}^{n} d_i \log(1/\epsilon))$ gradient computations, with no assumptions on the condition number. The previous best algorithm independent of the condition number is the standard cutting plane method, which requires $O(nd \log(1/\epsilon))$ gradient computations. As a corollary, we improve upon the evaluation oracle complexity for decomposable submodular minimization by [Axiotis, Karczmarz, Mukherjee, Sankowski and Vladu, ICML 2021]. Our main technical contribution is an adaptive procedure to select an $f_i$ term at every iteration via a novel combination of cutting-plane and interior-point methods.

---

[*]Authors ordered alphabetically.

36th Conference on Neural Information Processing Systems (NeurIPS 2022).

# 1   Introduction

Many fundamental problems in machine learning are abstractly captured by the convex optimization formulation

$$\text{minimize}_{\theta \in \mathbb{R}^d} \quad \sum_{i=1}^n f_i(\theta), \tag{1.1}$$

where each $f_i$ is a convex, Lipschitz function. For example, in empirical risk minimization, each $f_i$ measures the loss incurred by the $i$-th data point from the training set. In generalized linear models, each $f_i$ represents a link function applied to a linear predictor evaluated at the $i$-th data point.

The ubiquity of (1.1) in the setting with smooth $f_i$'s has spurred the development of well-known variants of stochastic gradient methods [44, 7, 55, 6] such as [46, 49, 25, 35, 15, 36, 3, 22, 47]; almost universally, these algorithms leverage the "sum structure" of (1.1) by sampling, in each iteration, one $f_i$ with which to make progress. These theoretical developments have in turn powered tremendous empirical success in machine learning through widely used software packages such as libSVM [13].

In many practical applications, (1.1) appears with non-smooth $f_i$'s, as well as the additional structure that each $f_i$ depends only on a subset of the problem parameters $\theta$. One notable example is decomposable submodular function minimization[2] (SFM), which has proven to be expressive in diverse contexts such as determinantal point processes [30], MAP inference in computer vision [29, 53, 17], hypergraph cuts [52], and covering functions [50]. Another application is found in generalized linear models when the data is high dimensional and sparse. In this setting, $f_i$ depends on a restricted subset of the parameters $\theta$ that correspond to the features of the data point with non-zero value. Last but not least, the case with each $f_i$ depending on a small subset of the parameters is also called sparse separable optimization and has applications in sparse SVM and matrix completion [42].

In this work, we initiate a systematic study of algorithms for (1.1) without the smoothness assumption[3]. Motivated by the aforementioned applications, we introduce the additional structure that each $f_i$ depends on a subset of the coordinates of $\theta$. As is standard in the black-box model for studying first-order convex optimization methods, we allow sub-gradient oracle access to each $f_i$.

**Problem 1.** *Let $f_1, f_2, \ldots, f_n : \mathbb{R}^d \mapsto \mathbb{R}$ be convex, L-Lipschitz, and possibly non-smooth functions, where each $f_i$ depends on $d_i$ coordinates of $\theta$, and is accessible via a (sub-)gradient oracle. Define $m := \sum_{i=1}^n d_i$ to be the "total effective dimension" of the problem. Let $\theta^\star := \arg\min_{\theta \in \mathbb{R}^d} \sum_{i=1}^n f_i(\theta)$ be a minimizer of (1.1), and let $\theta^{(0)}$ be an initial point such that $\|\theta^{(0)} - \theta^\star\|_2 \leq R$. We want to compute a vector $\theta \in \mathbb{R}^d$ satisfying*

$$\sum_{i=1}^n f_i(\theta) \leq \epsilon LR + \sum_{i=1}^n f_i(\theta^\star). \tag{1.2}$$

**Prior works.**   We focus on the weakly-polynomial regime and therefore restrict ourselves to algorithms with $\text{polylog}(1/\epsilon)$ gradient oracle complexities. Table 1 summarizes the performance of all well-known algorithms applied to Problem 1. Note that the variants of gradient descent each require bounded condition number. The results of [39, 2] and cutting plane methods are all complemented by matching lower bounds [54, 40]. Additionally, the work on non-smooth ERM crucially requires the objective function to be a sum of a smooth ERM part and a non-smooth regularizer. There are many important problems for which the objective function cannot be split in this way, so for these problems, the techniques developed for non-smooth ERM do not apply. In comparison, our work can be understood as dealing with a more general ERM problem with fewer structural assumptions.

Even with smooth $f_i$'s, first-order methods perform poorly when the condition number is large, or when there is a long chain of variable dependencies. These instances commonly arise in applications; an example from signal processing is

$$\text{minimize}_\mathbf{x} \left\{ (\mathbf{x}_1 - 1)^2 + \sum_{i=2}^{n-1} (\mathbf{x}_i - \mathbf{x}_{i+1})^2 + \mathbf{x}_n^2 \right\}, \tag{1.3}$$

---

[2]In decomposable submodular minimization, each $f_i$ corresponds to the Lovász extension of the individual submodular function and is therefore generally non-smooth.

[3]A function $f$ is said to be $\beta$-*smooth* if $f(y) \leq f(x) + \langle \nabla f(x), y - x \rangle + \beta/2 \|y - x\|_2^2$ for all $x, y$ and $\alpha$-*strongly-convex* if $f(y) \geq f(x) + \langle \nabla f(x), y - x \rangle + \alpha/2 \|y - x\|_2^2$ for all $x, y$. The condition number of $f$ is defined to be $\kappa = \beta/\alpha$.

| Authors | Algorithm Type | Gradient Queries | Non-smooth OK? |
|---|---|---|---|
| [12] | Gradient Descent (GD) | $O(n\kappa \log(1/\epsilon))$ | |
| [39] | Accelerated (Acc.) GD | $O(n\sqrt{\kappa} \log(1/\epsilon))$ | |
| [46, 49, 25] | Stochastic (Stoch.) Variance-Reduced GD | $O((n + \overline{\kappa}) \log(1/\epsilon))$ | |
| [48, 34, 18, 56, 1] | Acc. Stoch. Variance-Reduced GD | $O((n + \sqrt{n\overline{\kappa}}) \log(\overline{\kappa}) \log(1/\epsilon))$ | |
| [2] | Acc. Stoch. Variance-Reduced GD | $O((n + \sqrt{n\overline{\kappa}}) \log(1/\epsilon))$ | |
| [27, 37, 51, 5, 33, 24] | Cutting-Plane Method (CPM) | $O(nd \log(1/\epsilon))$ | ✓ |
| [31, 16] | Robust Interior-Point Method (IPM) | $O(\sum_{i=1}^{n} d_i^{3.5} \log(1/\epsilon))$ | ✓ |

Table 1: Gradient oracle complexities for solving (1.1) to $\epsilon$-additive accuracy. $\kappa$ denotes the condition number of $\sum_i f_i$, and $\overline{\kappa}$ is a variant of the condition number defined to be the sum of smoothness of the $f_i$'s divided by the strong convexity of $\sum_i f_i$.

whose variables form an $O(n)$-length chain of dependencies, and has condition number $\kappa = \Theta(n^2)$ and $\overline{\kappa} = \Theta(n^3)$. Gradient descent algorithms such as [39] and [2] therefore require $\Omega(n^2)$ gradient queries, despite the problem's total effective dimension being only $O(n)$.

On the other hand, cutting-plane methods (CPM) and robust interior-point methods (IPM) both trade off the dependency on condition number for worse dependencies on the problem dimension.

These significant gaps in the existing body of work motivate the following question:

> *Can we solve Problem 1 using a nearly-linear (in total effective dimension)*
> *number of sub-gradient oracle queries?*

In this paper, we give an affirmative answer to this question.

## 1.1  Our results

We present an algorithm to solve Problem 1 with gradient oracle complexity nearly-linear in the total effective dimension. Our main result is the following theorem.

**Theorem 1.2** (Main Result). *Given Problem 1 and $\theta^{(0)}$ such that $\|\theta^\star - \theta^{(0)}\|_2 \leq R$. Assuming all the $f_i$'s are L-Lipschitz, then there is an algorithm that in time $\mathrm{poly}(m \log(1/\epsilon))$, using $O(m \log(m/\epsilon))$ gradient oracle calls, outputs a vector $\theta \in \mathbb{R}^d$ such that*

$$\sum_{i=1}^{n} f_i(\theta) \leq \sum_{i=1}^{n} f_i(\theta^\star) + \epsilon \cdot LR.$$

Intuitively, the number of gradient queries for each $f_i$ should be thought of as $\widetilde{O}(d_i)$ in our algorithm, which nearly matches that of the standard cutting-plane method for minimizing the individual function $f_i$. The nearly-linear dependence on $m$ overall is obtained by leveraging the additional structure on the $f_i$'s and stands in stark contrast to the $O(nd)$ query complexity of CPM, which is significantly worse in the case where each $d_i \ll d$. Furthermore, we improve over the current best gradient descent algorithms in the case where the $f_i$'s have a large condition number.

Based on the query complexity of the standard cutting-plane method, we have the following lower bound matching our algorithm's query complexity up to a $\log m$-factor:

**Theorem 1.3.** *There exist functions $f_1, \ldots, f_n : \mathbb{R}^d \mapsto \mathbb{R}$ for which a total of $\Omega(m \log(1/\epsilon))$ gradient queries are required to solve Problem 1.*

An immediate application of Theorem 1.2 is to decomposable submodular function minimization:

**Theorem 1.4** (Decomposable SFM). *Let $V = \{1, 2, \ldots, n\}$, and $F : 2^V \mapsto [-1, 1]$ be given by $F(S) = \sum_{i=1}^{n} F_i(S \cap V_i)$, where each $F_i : 2^{V_i} \mapsto \mathbb{R}$ is a submodular function on $V_i \subseteq V$ with $|V_i| \leq k$. We can find an $\epsilon$-additive approximate minimizer of $F$ in $O(nk^2 \log(nk/\epsilon))$ evaluation oracle calls.*

Theorem 1.4 significantly improves over the evaluation oracle complexity of $\widetilde{O}(nk^6 \log(1/\epsilon))$ given in [4] when the dimension $k$ of each function $F_i$ is large. For non-decomposable SFM, i.e. $n = 1$ and

$|V_1| = k$, the current best weakly-polynomial time SFM algorithm[4] finds an $\epsilon$-approximate minimizer in time $O(k^2 \log(k/\epsilon))$ [33]. Therefore, our result in Theorem 1.4 can be viewed as a generalization of the evaluation oracle complexity for non-decomposable SFM in [33], and the dependence on $k$ in Theorem 1.4 might be the best possible. We defer the details of decomposable SFM to Appendix D.

### 1.1.1 Limitations

Some limitations of our algorithm are as follows: When each $f_i$ depends on the entire $d$-dimensional vector $\theta$, as opposed to a subset of the coordinates of size $d_i \ll d$, our gradient complexity simply matches that of CPM. We would like to highlight, though, that our focus is in fact the regime $d_i \ll d$. When the $f_i$'s are strongly-convex and smooth, our gradient complexity improves over Table 1 only when $\kappa$ is large compared to $d_i$. Finally, note that we consider only the gradient oracle complexity in our work; our algorithm's implementation requires sampling a Hessian matrix and a gradient vector at every iteration, which incur an additional $\text{poly}(m, \log(1/\epsilon))$ factor in the overall running time.

## 1.2 Technical challenges in prior works

We now describe the key technical challenges that barred existing algorithms from solving Problem 1 in the desired nearly-linear gradient complexity.

**Gradient descent and variants.** As mentioned in Section 1, the family of gradient descent algorithms presented in Table 1 are not applicable to Problem 1 without the smoothness assumption. When the objective in Problem 1 is smooth but has a large condition number, even the optimal deterministic algorithm, Accelerated Gradient Descent (AGD) [39] can perform poorly. For example, when applied to (1.3), AGD updates only one coordinate in each step (thereby requiring $n$ steps), with each step performing $n$ gradient queries (one on each term in the problem objective), yielding a total gradient complexity of $\Omega(n^2)$ [39]. For a similar reason, the fastest randomized algorithm, Katyusha [2] also incurs a gradient complexity of $\Omega(n^2)$ [54].

**Cutting-plane methods (CPM).** Given a convex function $f$ with its set $\mathcal{S}$ of minimizers, CPM minimizes $f$ by maintaining a convex search set $\mathcal{E}^{(k)} \supseteq \mathcal{S}$ in the $k^{\text{th}}$ iteration, and iteratively shrinking $\mathcal{E}^{(k)}$ using the sub-gradients of $f$. Specifically, this is achieved by noting that for any $\mathbf{x}^{(k)}$ chosen from $\mathcal{E}^{(k)}$, if the gradient oracle indicates $\nabla f(\mathbf{x}^{(k)}) \neq 0$, (i.e. $\mathbf{x}^{(k)} \notin \mathcal{S}$), then the convexity of $f$ guarantees $\mathcal{S} \subseteq \mathcal{H}^{(k)} := \{\mathbf{y} : \langle \nabla f(\mathbf{x}^{(k)}), \mathbf{y} - \mathbf{x}^{(k)} \rangle \leq 0\}$, and hence $\mathcal{S} \subseteq \mathcal{H}^{(k)} \cap \mathcal{E}^{(k)}$. The algorithm continues by choosing $\mathcal{E}^{(k+1)} \supseteq \mathcal{E}^{(k)} \cap \mathcal{H}^{(k)}$, and different choices of $\mathbf{x}^{(k)}$ and $\mathcal{E}^{(k)}$ yield different rates of shrinkage of $\mathcal{E}^{(k)}$ until a point in $\mathcal{S}$ is found.

Solving Problem 1 via the current fastest CPM [24] takes $\widetilde{O}(d)$ iterations, each invoking the gradient oracle on every $f_i$ to compute $\nabla f(\mathbf{x}^{(k)}) = \sum_{i=1}^n \nabla f_i(\mathbf{x}^{(k)})$. This results in $\widetilde{O}(nd)$ gradient queries overall, which can be quadratic in $n$ when $d = \Theta(n)$ even if each $f_i$ depends on only $d_i = O(1)$ coordinates. Similar to gradient descent and its variants, the poor performance of CPM on Problem 1 may therefore be attributed to their inability to query the right $f_i$ required to make progress.

**Interior-point methods (IPM).** IPM solves the convex program $\min_{\mathbf{u} \in \mathcal{S}} \langle \mathbf{c}, \mathbf{u} \rangle$ by solving a sequence of unconstrained problems $\min_{\mathbf{u}} \Psi_t(\mathbf{u}) := \{t \cdot \langle \mathbf{c}, \mathbf{u} \rangle + \psi_{\mathcal{S}}(\mathbf{u})\}$ parametrized by increasing $t$, where $\psi_{\mathcal{S}}$ is a *self-concordant barrier* function that enforces feasibility by becoming unbounded as it approaches the boundary of the feasible set $\mathcal{S}$. The algorithm starts at $t = 0$, for which an approximate minimizer $\mathbf{x}_0^\star$ of $\psi_{\mathcal{S}}$ is known, and alternates between increasing $t$ and updating to an approximate minimizer $\mathbf{x}_t^\star$ of the new $\Psi_t$ via Newton's method. For a sufficiently large $t$, the minimizer $\mathbf{x}_t^\star$ also approximately optimizes the original problem $\min_{\mathbf{u} \in \mathcal{S}} \langle \mathbf{c}, \mathbf{u} \rangle$ with sub-optimality gap $O(\nu/t)$, where $\nu$ is the self-concordance parameter of the barrier function used.

To apply IPM to Problem 1, we may first transform (1.1) to $\min_{(\mathbf{u}, \mathbf{z}) \in \mathcal{K}} \sum_i \mathbf{z}_i$, where $\mathcal{K} = \{(\mathbf{u}, \mathbf{z}) : (\mathbf{u}_i, \mathbf{z}_i) \in \mathcal{K}_i, \forall i \in [n]\}$ and $\mathcal{K}_i = \{(\mathbf{u}_i, \mathbf{z}_i) : f_i(\mathbf{u}_i) \leq \mathbf{z}_i\}$ is the feasible set. Using the universal barrier $\psi_i$ for each $\mathcal{K}_i$ [38], the number of iterations of IPM is $\widetilde{O}(\sqrt{\sum_{i=1}^n d_i})$, each requiring the computation of the Hessian and gradient of $\psi_i$ for all $i \in [n]$, leading to a total of $\widetilde{O}(n^{1.5})$ sub-gradient queries to $f_i$'s even when all $d_i = O(1)$. Even when leveraging the recent framework of

---

[4] Here, we focus on the weakly-polynomial regime, where the runtime dependence on $\epsilon$ is $\log(1/\epsilon)$.

robust IPM for linear programs [31], the computation of each Hessian (by sampling the corresponding $\mathcal{K}_i$ [23]) yields a total sub-gradient oracle complexity of $\widetilde{O}(\sum_{i=1}^{n} d_i^{3.5})$, far from the complexity we seek.

## 1.3 Our algorithmic framework

We now give an overview of the techniques developed in this work to overcome the above barriers. First, by making identical copies of coordinates shared by different $f_i$ and using the convex sets $\mathcal{K}_i$ to make the objective function linear, we transform (1.1) into a convex program over structured convex sets:

$$\begin{aligned}
\text{minimize} \quad & \langle \mathbf{c}, \mathbf{x} \rangle, \\
\text{subject to} \quad & \mathbf{x}_i \in \mathcal{K}_i \subseteq \mathbb{R}^{d_i+1} \ \forall i \in [n] \\
& \mathbf{Ax} = \mathbf{b}.
\end{aligned} \tag{1.4}$$

where $\mathbf{x}$ is the concatenation of the vectors $\mathbf{x}_1, \ldots, \mathbf{x}_n$, the objective vector $\mathbf{c}$ is 1 for the last coordinate of each $\mathbf{x}_i$ and 0 otherwise, and $\mathbf{Ax} = \mathbf{b}$ enforces that different copies of the same coordinates should be the same. Note that the sub-gradient oracle for $f_i$ can be transformed equivalently to a separation oracle $\mathcal{K}_i$. We define $\mathcal{K} := \mathcal{K}_1 \times \mathcal{K}_2 \times \ldots \times \mathcal{K}_n$.

**Main idea: combining CPM and IPM.** Recall that CPM maintains a convex set which initially contains the feasible region and gradually shrinks around the minimizer, while IPM maintains a point inside the feasible region that moves toward the minimizer. Our novel idea is to combine both methods and maintain *an inner convex set $\mathcal{K}_{in,i}$ as well as an outer convex set $\mathcal{K}_{out,i}$* for each $i \in [n]$, such that $\mathcal{K}_{in,i} \subseteq \mathcal{K}_i \subseteq \mathcal{K}_{out,i}$. We define $\mathcal{K}_{in}$ and $\mathcal{K}_{out}$ analogously to $\mathcal{K}$. When Inequality (3.4) and Inequality (3.3) are satisfied for all $i \in [n]$, we make IPM-style updates without needing to make any oracle calls. When Inequality (3.3) is violated for some $i \in [n]$, we query the separation oracle at the point $\mathbf{x}_{out,i}^\star$ defined as the centroid of $\mathcal{K}_{out,i}$ (c.f. Proposition 3.2). Based on the oracle's response, we iteratively either grow $\mathcal{K}_{in,i}$ (and, thus, $\mathcal{K}_{in}$) outward or shrink $\mathcal{K}_{out,i}$ (and, thus, $\mathcal{K}_{out}$) inward, until ultimately they approximate $\mathcal{K}$ around the optimum point.

**First benefit: large change in volume.** If the point $\mathbf{x}_{out,i}^\star$ violates Inequality (3.3) for some $i \in [n]$, we query the separation oracle to see if $\mathbf{x}_{out,i}^\star \in \mathcal{K}_i$ or not. If $\mathbf{x}_{out,i}^\star \in \mathcal{K}_i$, then it is used to expand $\mathcal{K}_{in,i}$, yielding in a large volume increase for $\mathcal{K}_{in,i}$. On the other hand, if $\mathbf{x}_{out,i}^\star \notin \mathcal{K}_i$, the fact that it is the centroid of $\mathcal{K}_{out,i}$ results in a large volume decrease for $\mathcal{K}_{out,i}$ when it is intersected with a halfspace through $\mathbf{x}_{out,i}^\star$. Thus, our algorithm witnesses a large change in volume of one of $\mathcal{K}_{in,i}$ and $\mathcal{K}_{out,i}$, regardless of the answer from the oracle. Just like in standard CPM, this rapid change in volume is crucial to achieving the algorithm's oracle complexity.

**Second benefit: making a smart choice about querying $f_i$.** Since the algorithm maintains both an inner and outer set approximating $\mathcal{K}$, by checking if $\mathcal{K}_{in,i}$ and $\mathcal{K}_{out,i}$ differ significantly (Inequality (3.3) essentially performs this function), we can determine if $\mathcal{K}_i$ is poorly approximated, and if so, improve the inner and outer approximations of the true feasible set. Choosing the right $\mathcal{K}_i$ translates to choosing the right $f_i$ to make progress with at an iteration; thus, we address the central weakness of the gradient descent variants in solving (1.1).

## 2 Notation and preliminaries

The full version of the preliminaries required to understand all proof details is in Appendix A.

We use lowercase boldface letters to denote (column) vectors and uppercase boldface letters to denote matrices. We use $\mathbf{x}_i$ to denote the $i^{\text{th}}$ block of coordinates in the vector $\mathbf{x}$ (the ordering of these blocks is not important in our setup).

We use $\langle \mathbf{x}, \mathbf{y} \rangle$ to mean the Euclidean inner product $\mathbf{x}^\top \mathbf{y}$. A subscript $\mathbf{x}$ in the inner product notation means it is induced by the Hessian of some function (which is locally specified) at $\mathbf{x}$; for example, $\langle \mathbf{u}, \mathbf{v} \rangle_{\mathbf{x}} = \mathbf{u}^\top \nabla_{ii}^2 \psi(\mathbf{x}) \mathbf{v}$ with $\psi$ separately specified. We define the local norm of $\mathbf{v}$ at $\mathbf{x}$ analogously: $\|\mathbf{v}\|_{\mathbf{x}} = \sqrt{\langle \mathbf{v}, \nabla^2 \psi(\mathbf{x}) \cdot \mathbf{v} \rangle}$. We also define the norm $\|\mathbf{v}\|_{\mathbf{x},1} := \sum_{i=1}^{n} \|\mathbf{v}\|_{\mathbf{x}_i}$.

We use $\psi$ to represent barrier functions and $\Phi$ to represent potential functions, with appropriate subscripts and superscripts to qualify them as needed.

**Definition 2.1** (Polar of a Set). Given a set $\mathcal{S} \subseteq \mathbb{R}^n$, its polar is defined as

$$\mathcal{S}^\circ := \{\mathbf{y} \in \mathbb{R}^n : \langle \mathbf{y}, \mathbf{x} \rangle \leq 1, \ \forall \mathbf{x} \in \mathcal{S}\}.$$

**Lemma 2.2** ([45]). *Let $\mathcal{S} \subseteq \mathbb{R}^n$ be a closed, compact, convex set, and let $y$ be a point. Then $(\text{conv}\{\mathcal{S}, \mathbf{y}\})^\circ \subseteq \mathcal{S}^\circ \cap \mathcal{H}$, where $\mathcal{H}$ is the halfspace defined by $\mathcal{H} = \{\mathbf{z} \in \mathbb{R}^n : \langle \mathbf{z}, \mathbf{y} \rangle \leq 1\}$.*

**Theorem 2.3** ([20, 11]). *Let $f$ be a log-concave distribution on $\mathbb{R}^d$ with centroid $\mathbf{c}_f$. Let $\mathcal{H} = \{\mathbf{u} \in \mathbb{R}^d : \langle \mathbf{u}, \mathbf{v} \rangle \geq q\}$ be a halfspace defined by a normal vector $\mathbf{v} \in \mathbb{R}^d$. Then, $\int_\mathcal{H} f(\mathbf{z})d\mathbf{z} \geq \frac{1}{e} - t^+$, where $t = \frac{q - \langle \mathbf{c}_f, \mathbf{v} \rangle}{\sqrt{\mathbb{E}_{\mathbf{y} \sim f} \langle \mathbf{v}, \mathbf{y} - \mathbf{c}_f \rangle^2}}$ is the distance of the centroid to the halfspace scaled by the standard deviation along the normal vector $\mathbf{v}$ and $t^+ := \max\{0, t\}$.*

**Definition 2.4** (Self-concordance). We call $F : Q \mapsto \mathbb{R}$ a self-concordant function on a convex set $Q$ if for any $\mathbf{x} \in Q$ and any $\mathbf{h}$,

$$|D^3 F(\mathbf{x})[\mathbf{h}, \mathbf{h}, \mathbf{h}]| \leq 2(D^2 F(\mathbf{x})[\mathbf{h}, \mathbf{h}])^{3/2},$$

where $D^k F(\mathbf{x})[\mathbf{h}_1, \ldots, \mathbf{h}_k]$ is the $k$-th derivative of $F$ at $\mathbf{x}$ along the directions $\mathbf{h}_1, \ldots, \mathbf{h}_k$. Additionally, if for any $\mathbf{x} \in Q$, we have $\nabla F(x)^\top (\nabla^2 F(x))^{-1} \nabla F(x) \leq \nu$, then $F$ is a $\nu$-self-concordant barrier.

Finally, we need the following definitions of entropic barrier and universal barrier.

**Definition 2.5** ([10, 14]). Given a convex body $\mathcal{K} \subseteq \mathbb{R}^n$ and some fixed $\theta \in \mathbb{R}^n$, define the function $f(\theta) = \log\left[\int_{\mathbf{x} \in \mathcal{K}} \exp\langle \mathbf{x}, \theta \rangle d\mathbf{x}\right]$. Then the Fenchel conjugate $f^* : \text{int}(\mathcal{K}) \to \mathbb{R}$ is a self-concordant barrier termed the *entropic barrier*. The entropic barrier is $n$-self-concordant.

**Definition 2.6** ([41, 32]). Given a convex body $\mathcal{K} \subseteq \mathbb{R}^n$, the *universal barrier* of $\mathcal{K}$ is defined as $\psi : \text{int}(\mathcal{K}) \to \mathbb{R}$ by

$$\psi(\mathbf{x}) = \log \text{vol}((\mathcal{K} - \mathbf{x})^\circ).$$

The universal barrier is $n$-self-concordant.

## 3  Our algorithm

We begin by reducing Problem 1 to the following slightly stronger formulation (see Theorem 1.2 for the detailed reduction):

$$\begin{aligned}
\text{minimize} \quad & \langle \mathbf{c}, \mathbf{x} \rangle, \\
\text{subject to} \quad & \mathbf{x}_i \in \mathcal{K}_i \subseteq \mathbb{R}^{d_i+1} \ \forall i \in [n] \\
& \mathbf{A}\mathbf{x} = \mathbf{b}.
\end{aligned} \tag{3.1}$$

where $\mathbf{x}$ is a concatenation of vectors $\mathbf{x}_i$'s, and the $\mathcal{K}_i$'s are disjoint convex sets. This formulation decouples the overlapping support of the original $f_i$'s by introducing additional variables tied together through the linear system $\mathbf{A}\mathbf{x} = \mathbf{b}$. Each $\mathcal{K}_i$ is constructed by applying a standard epigraph trick to the function $f_i$.

Note that we do not have a closed-form expression for $\mathcal{K}_i$. Instead, the subgradient oracle for $f_i$ translates to a *separation oracle* for $\mathcal{K}_i$: on a point $\mathbf{z}_i$ queried by the oracle, the oracle either asserts $\mathbf{z}_i \in \mathcal{K}_i$, or returns a separating hyperplane that separates $\mathbf{z}_i$ from $\mathcal{K}_i$.

At the start of our algorithm, we have the following guarantee:

**Lemma 3.1.** *At the start of our algorithm, we are guaranteed the existence of the following.*

*1. Explicit convex sets $\mathcal{K}_{in} := \mathcal{K}_{in,1} \times \mathcal{K}_{in,2} \times \cdots \times \mathcal{K}_{in,n}$ and $\mathcal{K}_{out} := \mathcal{K}_{out,1} \times \mathcal{K}_{out,2} \times \cdots \times \mathcal{K}_{out,n}$ such that $\mathcal{K}_{in} \subseteq \mathcal{K} := \mathcal{K}_1 \times \cdots \times \mathcal{K}_n \subseteq \mathcal{K}_{out}$,*

*2. An initial $\mathbf{x}_{initial} \in \mathcal{K}_{in}$ such that $\mathbf{A}\mathbf{x}_{initial} = \mathbf{b}$.*

We show how to construct such a set $\mathcal{K}_{in}$ in Appendix C.1 and how to find such a $\mathcal{K}_{out}$ and $\mathbf{x}_{initial}$ in Appendix C.2.

---

**Algorithm 1** Minimizing Decomposable Convex Function

---
1: ▷ Solving Problem 3.1
2: **Input.** $\epsilon$, $\mathbf{A}$, $\mathbf{b}$, $\mathbf{c}$, $R$, $r$, $m$, $n$, $\mathbf{x}$, $\mathcal{K}_{\text{in}}$, $\mathcal{K}_{\text{out}}$, and $\mathcal{O}_i$ for each $i \in [n]$.
3: **Initialization.** $t = \frac{m \log m}{\sqrt{n}\|\mathbf{c}\|_2 R}$ and $t_{\text{end}} = \frac{8m}{\epsilon \|\mathbf{c}\|_2 R}$, $\eta = \frac{1}{100}$, and $\mathbf{x}^{\star}_{\text{out}}$ (via Equation (3.2))
4: **while** true **do**
5:     **if** $\langle \mathbf{c}, \mathbf{x} \rangle \leq \langle \mathbf{c}, \mathbf{x}^{\star}_{\text{out}} \rangle + \frac{4m}{t}$ **then**            ▷ Either update $t$ or end the algorithm
6:         **if** $t \geq t_{\text{end}}$ **then**
7:             **return** $\arg\min_{\mathbf{x}:\mathbf{x}\in\mathcal{K}_{\text{in}}, \mathbf{A}\mathbf{x}=\mathbf{b}} \{t\langle\mathbf{c},\mathbf{x}\rangle + \sum_{i=1}^{n} \psi_{\text{in},i}(\mathbf{x}_i)\}$.     ▷ End the algorithm
8:         **end if**
9:         $t \leftarrow t \cdot \left[1 + \frac{\eta}{4m}\right]$                                ▷ Update $t$
10:         Update $\mathbf{x}^{\star}_{\text{out}}$ and jump to Line 4       ▷ $\mathbf{x}^{\star}_{\text{out}}$ computed as as per Equation (3.2)
11:     **end if**
12:     **for** all $i \in [n]$ **do**
13:         **if** $\langle \nabla \psi_{\text{in},i}(\mathbf{x}_i), \mathbf{x}^{\star}_{\text{out},i} - \mathbf{x}_i \rangle + \eta \|\mathbf{x}^{\star}_{\text{out},i} - \mathbf{x}_i\|_{\mathbf{x}_i} \geq 4d_i$ **then**
14:             **if** $\mathbf{x}^{\star}_{\text{out},i} \in \mathcal{K}_i$ **then**                         ▷ Query $\mathcal{O}_i$
15:                 $\mathcal{K}_{\text{in},i} = \text{conv}(\mathcal{K}_{\text{in},i}, \mathbf{x}^{\star}_{\text{out},i})$             ▷ Update $\mathcal{K}_{\text{in},i}$
16:             **else**
17:                 $\mathcal{K}_{\text{out},i} = \mathcal{K}_{\text{out},i} \cap \mathcal{H}_i$, where $\mathcal{H}_i = \mathcal{O}_i(\mathbf{x}^{\star}_{\text{out},i})$     ▷ Update $\mathcal{K}_{\text{out},i}$
18:             **end if**
19:             Update $\mathbf{x}^{\star}_{\text{out}}$ and jump to Line 4       ▷ $\mathbf{x}^{\star}_{\text{out}}$ computed as per Equation (3.2)
20:         **end if**
21:     **end for**
22:     Set $\delta_{\mathbf{x}} := \frac{\eta}{2} \cdot \frac{\mathbf{x}^{\star}_{\text{out}} - \mathbf{x}}{\|\mathbf{x}^{\star}_{\text{out}} - \mathbf{x}\|_{\mathbf{x},1}}$, where $\|\mathbf{u}\|_{\mathbf{x},1} := \sum_{i=1}^{n} \|\mathbf{u}\|_{\mathbf{x}_i}$.
23:     $\mathbf{x} \leftarrow \mathbf{x} + \delta_{\mathbf{x}}$                                  ▷ Move $\mathbf{x}$ towards $\mathbf{x}^{\star}_{\text{out}}$
24: **end while**

---

## 3.1 Details of our algorithm

In this section, we explain our main algorithm (Algorithm 1).

The inputs to Algorithm 1 are: initial sets $\mathcal{K}_{\text{in}}$ and $\mathcal{K}_{\text{out}}$ satisfying $\mathcal{K}_{\text{in}} \subseteq \mathcal{K} \subseteq \mathcal{K}_{\text{out}}$, an initial point $\mathbf{x} \in \mathcal{K}_{\text{in}}$ satisfying $\mathbf{A}\mathbf{x} = \mathbf{b}$, a separation oracle $\mathcal{O}_i$ for each $\mathcal{K}_i$, the objective vector $\mathbf{c}$, and scalar parameters $m$, $n$, $R$, $r$, and $\epsilon$. All the parameters are set in the proof of Theorem B.10.

Throughout the algorithm, we maintain a central path parameter $t$ for IPM-inspired updates, the current solution $\mathbf{x}$, and convex sets $\mathcal{K}_{\text{in},i}$ and $\mathcal{K}_{\text{out},i}$ satisfying $\mathcal{K}_{\text{in},i} \subseteq \mathcal{K}_i \subseteq \mathcal{K}_{\text{out},i}$ for each $i \in [n]$. To run IPM-style updates, we choose the entropic barrier on $\mathcal{K}_{\text{out}}$ and the universal barrier on $\mathcal{K}_{\text{in}}$.

Given the current set $\mathcal{K}_{\text{out}}$, the current $t$, and the entropic barrier $\psi_{\text{out}}$ defined on $\widehat{\mathcal{K}}_{\text{out}} := \mathcal{K}_{\text{out}} \cap \{\mathbf{u} : \mathbf{A}\mathbf{u} = \mathbf{b}\}$, we define the point

$$\mathbf{x}^{\star}_{\text{out}} := \arg \min_{\mathbf{x} \in \widehat{\mathcal{K}}_{\text{out}}} \{t\langle \mathbf{c}, \mathbf{x} \rangle + \psi_{\text{out}}(\mathbf{x})\}. \tag{3.2}$$

Per the IPM paradigm, for the current value of $t$, this point serves as a target to "chase" when optimizing $\langle \mathbf{c}, \mathbf{x} \rangle$ over the set $\widehat{\mathcal{K}}_{\text{out}}$. Although our overall goal in Problem 3.1 is to optimize over $\mathcal{K} \cap \{\mathbf{u} : \mathbf{A}\mathbf{u} = \mathbf{b}\}$, we do not know $\mathcal{K}$ explicitly and therefore must use its known proxies, $\mathcal{K}_{\text{in}}$ or $\mathcal{K}_{\text{out}}$; we choose $\mathcal{K}_{\text{out}}$ because $\mathcal{K}_{\text{out}} \supseteq \mathcal{K}$ ensures we do not miss a potential optimal point.

Having computed the current target $\mathbf{x}^{\star}_{\text{out}}$, we move the current solution $\mathbf{x}$ towards it by taking a Newton step, provided certain conditions of feasibility and minimum progress are satisfied. If either one of these conditions is violated, we update either $\mathcal{K}_{\text{in}}$, $\mathcal{K}_{\text{out}}$, or the parameter $t$.

**Updating x.** In order to move $\mathbf{x}$ towards $\mathbf{x}^{\star}_{\text{out}}$, we require two conditions to hold: $\mathbf{x}^{\star}_{\text{out}} \in \mathcal{K}_{\text{in}}$ and $\langle \mathbf{c}, \mathbf{x} \rangle \geq \langle \mathbf{c}, \mathbf{x}^{\star}_{\text{out}} \rangle + O(1/t)$.

The first condition implies $\mathbf{x}^{\star}_{\text{out}} \in \mathcal{K}$, which would in turn ensure feasibility of the resulting $\mathbf{x}$ after a Newton step. To formally check this condition, we check if the following inequality is satisfied for all $i \in [n]$ and for a fixed constant $\eta$:

$$\langle \nabla \psi_{\text{in},i}(\mathbf{x}_i), \mathbf{x}^{\star}_{\text{out},i} - \mathbf{x}_i \rangle + \eta \cdot \|\mathbf{x}^{\star}_{\text{out},i} - \mathbf{x}_i\|_{\mathbf{x}_i} \leq 4d_i. \tag{3.3}$$

The intuition is that since any point within the domain of a self-concordant barrier satisfies the inequalities in Theorem A.5 and Theorem A.6, violating Inequality (3.3) implies that $\mathbf{x}^\star_{\text{out},i}$ is far from $\mathcal{K}_{\text{in},i}$, and as a result, $\mathbf{x}^\star_{\text{out}}$ is not a good candidate to move $\mathbf{x}$ towards.

The second condition we impose, one of "sufficient suboptimality", ensures significant progress in the objective value can be made when updating $\mathbf{x}$. Formally, we check if

$$\mathbf{c}^\top \mathbf{x}^\star_{\text{out}} + \frac{4m}{t} \le \mathbf{c}^\top \mathbf{x}. \tag{3.4}$$

If the inequality holds, then there is still "room for progress" to lower the value of $\langle \mathbf{c}, \mathbf{x} \rangle$ by updating $\mathbf{x}$; if the inequality is violated, we update $t$ instead.

Given the two conditions hold, we move $\mathbf{x}$ towards $\mathbf{x}^\star_{\text{out}}$ in Line 23. The update step is normalized by the distance between $\mathbf{x}$ and $\mathbf{x}^\star_{\text{out}}$ measured in the local norm, which enforces $\mathbf{x} \in \mathcal{K}$ (since by the definition of self-concordance, the unit radius Dikin ball lies inside the domain of the self-concordance barrier), and also helps bound certain first-order error terms (Inequality (B.14) in Appendix B.3).

The rest of this section details the procedure for when either of these conditions is violated.

**Updating the inner and outer convex sets.** Suppose Inequality (3.3) is violated for some $i \in [n]$. Then $\mathbf{x}^\star_{\text{out},i} \notin \mathcal{K}_{\text{in},i}$, which in turn means $\mathbf{x}^\star_{\text{out}}$ *might* not be in the feasible set $\mathcal{K}$. To reestablish Inequality (3.3) for $i$, we can either update $\mathcal{K}_{\text{in},i}$, or update $\mathcal{K}_{\text{out},i}$ and compute a new $\mathbf{x}^\star_{\text{out},i}$ by Equation (3.2).

To decide which option to take, we query $\mathcal{O}_i$ at the point $\mathbf{x}^\star_{\text{out},i}$: if the oracle indicates that $\mathbf{x}^\star_{\text{out},i} \in \mathcal{K}_i$, then we incorporate $\mathbf{x}^\star_{\text{out},i}$ into $\mathcal{K}_{\text{in},i}$ by redefining $\mathcal{K}_{\text{in},i} = \text{conv}(\mathcal{K}_{\text{in},i}, \mathbf{x}^\star_{\text{out},i})$ to be the convex hull of the current $\mathcal{K}_{\text{in},i}$ and $\mathbf{x}^\star_{\text{out},i}$ (Line 15). If, on the other hand, $\mathbf{x}^\star_{\text{out},i} \notin \mathcal{K}_i$, the oracle $\mathcal{O}_i$ will return a halfspace $\mathcal{H}_i$ satisfying $\mathcal{H}_i \supseteq \mathcal{K}_i$. Then we redefine $\mathcal{K}_{\text{out},i} = \mathcal{K}_{\text{out},i} \cap \mathcal{H}_i$ (Line 17). After processing this update of the sets, the algorithm recomputes $\mathbf{x}^\star_{\text{out}}$ and returns to the main loop since updating the sets does not necessarily imply that the new $\mathbf{x}^\star_{\text{out}}$ satisfies $\mathbf{x}^\star_{\text{out}} \in \mathcal{K}_{\text{in}}$.

This update rule for the sets is exactly where our novelty lies: *we do not arbitrarily update sets, rather, we update one only after checking the very specific condition $\mathbf{x}^\star_{out,i} \notin \mathcal{K}_{in,i}$.* Since the separation oracle is called only in this part of the algorithm, performing this check enables us to dramatically reduce the number of calls we make to the separation oracle, thereby improving our oracle complexity.

Further, this update rule shows that even when we cannot update the current $\mathbf{x}$, we make progress by using all the information from the oracles. Over the course of the algorithm, we gradually expand $\mathcal{K}_{\text{in}}$ and shrink $\mathcal{K}_{\text{out}}$, until they well-approximate $\mathcal{K}$. To formally quantify the change in volume due to the above operations, we consider the following alternative view of $\mathbf{x}^\star_{\text{out}}$.

**Proposition 3.2** (Section 3 in [10]; Section 3 of [28])**.** *Let $\theta \in \mathbb{R}^n$, and let $p_\theta$ be defined as $p_\theta(\mathbf{x}) := \exp(\langle \theta, \mathbf{x} \rangle - f(\theta))$, where $f(\theta) := \log \left[ \int_{\mathcal{K}} \exp(\langle \theta, \mathbf{u} \rangle) d\mathbf{u} \right]$. Then,*

$$\mathbb{E}_{\mathbf{x} \sim p_\theta}[\mathbf{x}] = \arg \min_{\mathbf{x} \in \text{int}(\mathcal{K})} \left\{ f^*(\mathbf{x}) - \langle \theta, \mathbf{x} \rangle \right\}.$$

By this proposition, $\mathbf{x}^\star_{\text{out}}$ defined in Equation (3.2) satisfies

$$\mathbf{x}^\star_{\text{out}} := \mathbb{E}_{\mathbf{x} \sim \exp\left\{ -t\langle \mathbf{c}, \mathbf{x} \rangle - \log\left[ \int_{\widehat{\mathcal{K}}_{\text{out}}} \exp(-t\langle \mathbf{c}, \mathbf{u} \rangle) d\mathbf{u} \right] \right\}}[\mathbf{x}], \tag{3.5}$$

that is, $\mathbf{x}^\star_{\text{out}}$ is the centroid of some exponential distribution over $\widehat{\mathcal{K}}_{\text{out}}$. As a result, if $\mathbf{x}^\star_{\text{out},i} \notin \mathcal{K}_i$, the hyperplane cutting $\widehat{\mathcal{K}}_{\text{out}}$ through $\mathbf{x}^\star_{\text{out}}$ will yield a large decrease in volume of $\widehat{\mathcal{K}}_{\text{out}}$, per Remark 9. Therefore, the query result in a large change in volume in either $\mathcal{K}_{\text{in}}$ or $\mathcal{K}_{\text{out}}$, allowing us to approximate $\mathcal{K}$ with a bounded number of iterations.

**Updating $t$.** If Inequality (3.4) is violated, then the current $\mathbf{x}$ is "as optimal as one can get" for the current parameter $t$. This could mean one of two things:

The first possibility is that we have already reached an approximate optimum, which we verify by checking whether $t \ge O(1/\epsilon)$ in Line 6: If true, this indicates that we have attained our desired suboptimality, and the algorithm terminates by returning

$$\mathbf{x}^{\text{ret}} = \arg \min_{\mathbf{x}: \mathbf{x} \in \mathcal{K}_{\text{in}}, \mathbf{A}\mathbf{x} = \mathbf{b}} \left\{ t \cdot \langle \mathbf{c}, \mathbf{x} \rangle + \sum_{i=1}^{n} \psi_{\text{in},i}(\mathbf{x}_i) \right\}.$$

The point $\mathbf{x}^{\text{ret}}$ is feasible because it is in $\mathcal{K}_{\text{in}}$ by definition, and the suboptimality of $O(1/t_{\text{end}}) = O(\epsilon)$ ensures it is an approximate optimum for the original problem.

The second possibility is that we need to increase $t$ to set the next "target suboptimality". The value of $t$ is increased by a scaling factor of $1 + O(1/m)$ in Line 9. This scaling factor ensures, like in the standard IPM framework, that the next optimum is not too far from the current one. Following the update to $t$, we recompute $\mathbf{x}_{\text{out}}^{\star}$ by Equation (3.2). Since $\langle \mathbf{c}, \mathbf{x} \rangle > \langle \mathbf{c}, \mathbf{x}_{\text{out}}^{\star} \rangle + O(1/t)$ is not guaranteed with the new $t$ and $\mathbf{x}_{\text{out}}^{\star}$, the algorithm jumps back to the start of the main loop.

# 4 Our analysis

To analyze the oracle complexity of Algorithm 1, we define the following potential function that captures the changes in $\mathcal{K}_{\text{in}}$, $\mathcal{K}_{\text{out}}$, $t$, and $\mathbf{x}$ in each iteration:

$$\Phi := \underbrace{t\langle \mathbf{c}, \mathbf{x} \rangle + \log\left[\int_{\widehat{\mathcal{K}}_{\text{out}}} \exp(-t\langle \mathbf{c}, \mathbf{u} \rangle) d\mathbf{u}\right]}_{\text{entropic terms}} + \underbrace{\sum_{i \in [n]} \psi_{\text{in},i}(\mathbf{x}_i)}_{\text{universal terms}}, \tag{4.1}$$

where $\log\left[\int_{\widehat{\mathcal{K}}_{\text{out}}} \exp(-t\langle \mathbf{c}, \mathbf{u} \rangle) d\mathbf{u}\right]$ is related to the entropic barrier on $\widehat{\mathcal{K}}_{\text{out}}$ (see Appendix B.1) and $\psi_{\text{in}}$ is the universal barrier on $\mathcal{K}_{\text{in}}$. We have the following bounds on the change in potential when $\mathcal{K}_{\text{in},i}$, $\mathcal{K}_{\text{out},i}$, $t$, and $\mathbf{x}$ are updated respectively; the formal statements are found in Appendix B.1, Appendix B.2, and Appendix B.3.

**Lemma 4.1.** *Let* $\mathbf{c}$ *be a fixed vector. Let* $\Phi$ *be the potential given in Equation* (4.1)*, defined as a function of* $\mathcal{K}_{in}$*,* $\mathcal{K}_{out}$*,* $\mathbf{x}$*, and* $t$*. Following an update to one of the parameters while keeping the others fixed, we have the following bounds on the new potential* $\Phi^{(new)}$*:*

*1. Let* $\mathcal{H}_i$ *be the halfspace generated by the separation oracle* $\mathcal{O}_i$ *queried at* $\mathbf{x}_{out,i}^{\star}$ *as shown in Line 17 of Algorithm 1. Then the new potential* $\Phi^{(new)}$ *defined with respect to* $\widehat{\mathcal{K}}_{out} \cap \mathcal{H}_i$ *is bounded by*

$$\Phi^{(new)} \leq \Phi + \log(1 - 1/e).$$

*2. When* $t$ *is updated to* $t \cdot \left[1 + \frac{\eta}{4m}\right]$ *in Line 9 of Algorithm 1, the new potential* $\Phi^{(new)}$ *defined with respect to the updated* $t$ *is bounded by*

$$\Phi^{(new)} \leq \Phi + \eta + \eta^2.$$

*3. Given a convex set* $\mathcal{K} \subseteq \mathbb{R}^d$ *and a point* $\mathbf{x} \in \mathcal{K}$*, let* $\psi_{\mathcal{K}} := \log \text{vol}(\mathcal{K} - \mathbf{x})^{\circ}$ *be the universal barrier defined on* $\mathcal{K}$ *with respect to* $\mathbf{x}$*. For some fixed* $\eta \leq 1/4$*, let* $\mathbf{y} \in \mathcal{K}$ *be a point satisfying*

$$\langle \nabla\psi_{\mathcal{K}}(\mathbf{x}), \mathbf{y} - \mathbf{x} \rangle + \eta\|\mathbf{y} - \mathbf{x}\|_{\mathbf{x}} \geq 4d, \tag{4.2}$$

*and construct the new set* $\text{conv}\{\mathcal{K}, \mathbf{y}\}$*. Then, the value of the universal barrier on the new set with respect to* $\mathbf{x}$ *satisfies*

$$\psi_{\text{conv}(\mathcal{K}, \mathbf{y})}(\mathbf{x}) \leq \psi_{\mathcal{K}}(\mathbf{x}) + \log(1 - 1/e + \eta).$$

*4. Consider the update step* $\delta_{\mathbf{x}} = \frac{\eta}{2} \cdot \frac{\mathbf{x}_{out}^{\star} - \mathbf{x}}{\|\mathbf{x}_{out}^{\star} - \mathbf{x}\|_{\mathbf{x},1}}$ *as in Line 22. Assume the guarantees in Inequality (3.3) and Inequality (3.4). Then the new potential* $\Phi^{(new)}$ *defined with respect to the updated* $\mathbf{x}$ *is bounded by*

$$\Phi^{(new)} \leq \Phi - \frac{\eta^2}{4}.$$

*Proof sketch.* We use tools from convex analysis, self-concordance calculus, and Grünbaum's results to derive the above potential bounds, as explained below.

1. Consider the update of $\widehat{\mathcal{K}}_{\text{out}}$ to $\widehat{\mathcal{K}}_{\text{out}} \cap \mathcal{H}_i$, with $\mathcal{H}_i$ passing through $\mathbf{x}_{\text{out},i}^{\star}$. Recall from Equation (3.5), $\mathbf{x}_{\text{out}}^{\star}$ is the centroid of $\widehat{\mathcal{K}}_{\text{out}}$ with respect to the (log-concave) distribution $\exp\left\{-t\langle \mathbf{c}, \mathbf{x} \rangle - \log\left[\int_{\widehat{\mathcal{K}}_{\text{out}}} \exp(-t\langle \mathbf{c}, \mathbf{u} \rangle) d\mathbf{u}\right]\right\}$. Therefore, per Remark 9, the volume of $\widehat{\mathcal{K}}_{\text{out}} \cap \mathcal{H}_i$ is at most $1 - 1/e$ factor of the volume of the original $\widehat{\mathcal{K}}_{\text{out}}$, which translates to the claimed potential change.

2. The change in $\Phi$ by updating $t$ to $t \cdot (1 + h)$ for some $h > 0$ is

$$\Phi^{(\text{new})} - \Phi = \left\{ \log \left[ \int_{\widehat{\mathcal{K}}_{\text{out}}} \exp\langle -t(1 + h)\mathbf{c}, \mathbf{v}\rangle d\mathbf{v} \right] - \log \left[ \int_{\widehat{\mathcal{K}}_{\text{out}}} \exp\langle -t\mathbf{c}, \mathbf{v}\rangle d\mathbf{v} \right] \right\} + th \cdot \langle \mathbf{c}, \mathbf{x}\rangle.$$

(4.3)

To bound the right-hand side, the key observation is that by duality with respect to the entropic barrier (Definition 2.5), the function $\log \left[ \int_{\mathbf{x} \in \widehat{\mathcal{K}}_{\text{out}}} \exp(\langle \mathbf{x}, \theta\rangle)d\mathbf{x} \right]$ is also self-concordant. This in turn allows us to bound the difference between the first two terms in the right hand side of Equation (4.3).

3. Lemma 2.2 implies $(\text{conv}(\mathcal{K}, \mathbf{y}) - \mathbf{x})^\circ = (\mathcal{K} - \mathbf{x})^\circ \cap \mathcal{H}$, where $\mathcal{H} = \{\mathbf{z} \in \mathbb{R}^n : \langle \mathbf{z}, \mathbf{y} - \mathbf{x}\rangle \leq 1\}$. Therefore $\psi_{\text{conv}(\mathcal{K}, \mathbf{y})}(\mathbf{x}) - \psi_{\mathcal{K}}(\mathbf{x})$ may be bounded via the change in volume of a set after intersection with a halfspace, for which we can again apply Grünbaum's result (Theorem 2.3). To argue that the halfspace $\mathcal{H}$ cuts the set $(\mathcal{K} - \mathbf{x})^\circ$ at a point not too far from the centroid of $(\mathcal{K} - \mathbf{x})^\circ$, we use the intuition that any $\mathbf{y}$ satisfying Equation (4.2) must be far from $\mathcal{K}$, by properties of the self-concordant function $\psi_{\mathcal{K}}$. This means $\text{conv}(\mathcal{K}, \mathbf{y})$ is much larger than $\mathcal{K}$, giving the desired conclusion by duality.

4. The change in potential due to change in $\mathbf{x}$ as per the stated update rule is given by

$$\Phi^{(\text{new})} - \Phi = \langle \nabla_{\mathbf{x}}\Phi, \delta_{\mathbf{x}}\rangle + \sum_{i=1}^{n} \left[\psi_{\text{in},i}(\mathbf{x}_i + \delta_{\mathbf{x},i}) - \psi_{\text{in},i}(\mathbf{x}_i) - \langle \nabla\psi_{\text{in},i}(\mathbf{x}_i), \delta_{\mathbf{x},i}\rangle\right].$$

The term in the summation on the right-hand side measures the error due to the first-order approximation of $\psi_{\text{in},i}$ around $\mathbf{x}_i$, which can be bounded by self-concordance. To bound the remaining linear term, we use $\nabla_{\mathbf{x}}\Phi = t\mathbf{c} + \sum_{i=1}^{n} \nabla\psi_{\text{in},i}(\mathbf{x}_i)$ and the assumed inequalities. □

By adding all the potential changes shown in the above lemma and using the bounds we derived on lower and upper potentials (c.f. Lemma B.8), we can show (c.f. Lemma B.9) the oracle complexity of Algorithm 1 to be $O\left(m \log\left(\frac{mR}{\epsilon r}\right)\right)$. Combined with our results on initialization then yields the oracle complexity of Problem 3.1 (c.f. Theorem B.10) and that of Problem 1 (c.f. Theorem 1.2). The full details of this may be found in Appendix B.4.

## 4.1 A remark on implementability

The gradient and hessian of the entropy barrier are the centroid and covariance of some exponential distribution over a polytope. In our setting, we need only a constant factor approximation of these quantities. While sampling from a log-concave distribution in general takes $n^{4.2}$ time, in our case, sampling from an exponential distribution over an explicit sparse polytope is faster, taking around quadratic time in practice because of our assumption that each $f_i$ depends only on a few variables.

These results (and runtime complexity) of sampling from an exponential distribution over explicit polytope have practical motivation as well, for example, in simulating metabolic networks (which can contain hundreds of thousands of variables) in bioinformatics. As a result, there has been extensive work developing new software for this task. One such popular package is `https://opencobra.github.io/cobratoolbox/stable/`. It can sample from a polytope with tens of thousands of variables currently in seconds. This package includes ACHR, CHRR, RHMC, all representing three generations of improvement on the sampler in last 20 years. For example, for one human model (RECON3) with 12k variables, ACHR probably takes $10^8$ cpu-sec to obtain one sample from that polytope while CHRR takes $10^4$ cpu-sec and RHMC takes 30 sec.

Therefore, even though the focus of our work has been on developing the theory, the preceding discussion suggests that there could be an implementation of our algorithm which can perform reasonably well in practice.

## Acknowledgements

We thank the anonymous reviewers of NeurIPS 2022 for very helpful comments. Research of YL is supported by NSF awards CCF-1749609, DMS-1839116, DMS-2023166, CCF-2105772, a Microsoft Research Faculty Fellowship, a Sloan Research Fellowship, and a Packard Fellowship. The full version of this paper is available at `https://arxiv.org/pdf/2208.03811.pdf`.

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
