Our appendix is organized as follows. Appendix A lays out all the preliminaries for the proof details of the paper; Appendix B shows the analysis of the algorithm; Appendix C shows the initialization, and Appendix D contains details about submodular function minimization as it pertains to our problem.

## A    Preliminaries

### A.1    Facts from convex analysis

In this section, we present some definitions and properties from convex analysis that are useful in our paper. These results are standard and may be found in, for example, [45, 8].

**Definition A.1.** Let $f : \mathbb{R}^n \to \mathbb{R}$. Then the function $f^* : \mathbb{R}^n \to \mathbb{R}$ defined as

$$f^*(\mathbf{y}) = \sup_{\mathbf{x} \in \text{dom}(f)} [\langle \mathbf{x}, \mathbf{y} \rangle - f(\mathbf{x})]$$

is called the Fenchel conjugate of the function $f$. An immediate consequence of the definition (and by applying the appropriate convexity-preserving property) is that $f^*$ is convex, regardless of the convexity of $f$. We use the superscript $*$ on functions to denote their conjugates.

**Lemma A.2** (Biconjugacy). *For a closed, convex function $f$, we have $f = f^{**}$.*

**Lemma A.3** ([45]). *For a closed, convex differentiable function $f$, we have $\mathbf{y} = \nabla f(\mathbf{x}) \iff \mathbf{x} = \nabla f^*(\mathbf{y})$.*

**Lemma A.4** ([45]). *For a strictly convex, twice-differentiable function $f$, we have $\nabla^2 f^*(\nabla f(\mathbf{x})) = (\nabla^2 f(\mathbf{x}))^{-1}$.*

**Definition 2.1** (Polar of a Set). Given a set $\mathcal{S} \subseteq \mathbb{R}^n$, its polar is defined as

$$\mathcal{S}^\circ := \{\mathbf{y} \in \mathbb{R}^n : \langle \mathbf{y}, \mathbf{x} \rangle \leq 1, \ \forall \mathbf{x} \in \mathcal{S}\}.$$

**Lemma 2.2** ([45]). *Let $\mathcal{S} \subseteq \mathbb{R}^n$ be a closed, compact, convex set, and let $y$ be a point. Then $(\text{conv}\{\mathcal{S}, \mathbf{y}\})^\circ \subseteq \mathcal{S}^\circ \cap \mathcal{H}$, where $\mathcal{H}$ is the halfspace defined by $\mathcal{H} = \{\mathbf{z} \in \mathbb{R}^n : \langle \mathbf{z}, \mathbf{y} \rangle \leq 1\}$.*

### A.2    Background on interior-point methods

Our work draws heavily upon geometric properties of self-concordant functions, which underpin the rich theory of interior-point methods. We list below the formal results needed for our analysis, and refer the reader to [38, 43] for a detailed exposition of this function class. We begin with the definitions of self-concordant functions and self-concordant barriers:

**Definition 2.4** (Self-concordance). We call $F : Q \mapsto \mathbb{R}$ a self-concordant function on a convex set $Q$ if for any $\mathbf{x} \in Q$ and any $\mathbf{h}$,

$$|D^3 F(\mathbf{x})[\mathbf{h}, \mathbf{h}, \mathbf{h}]| \leq 2(D^2 F(\mathbf{x})[\mathbf{h}, \mathbf{h}])^{3/2},$$

where $D^k F(\mathbf{x})[\mathbf{h}_1, \ldots, \mathbf{h}_k]$ is the $k$-th derivative of $F$ at $\mathbf{x}$ along the directions $\mathbf{h}_1, \ldots, \mathbf{h}_k$. Additionally, if for any $\mathbf{x} \in Q$, we have $\nabla F(x)^\top (\nabla^2 F(x))^{-1} \nabla F(x) \leq \nu$, then $F$ is a $\nu$-self-concordant barrier.

**Theorem A.5** (Theorem 2.3.3 from [43]). *If $f$ is a self-concordant barrier, then for all $\mathbf{x}$ and $\mathbf{y} \in dom(f)$, we have $\langle \nabla f(\mathbf{x}), \mathbf{y} - \mathbf{x} \rangle \leq \nu$, where $\nu$ is the self-concordance of $f$.*

**Theorem A.6** (Theorem 2.3.4 from [43]). *If $f$ is a $\nu$-self-concordant barrier such that $\mathbf{x}, \mathbf{y} \in dom(f)$ satisfy $\langle \nabla f(\mathbf{x}), \mathbf{y} - \mathbf{x} \rangle \geq 0$, then $\mathbf{y} \in \mathcal{B}_\mathbf{x}(\mathbf{x}, 4\nu + 1)$.*

We now state the following result from self-concordance calculus.

**Theorem A.7** (Theorem 3.3.1 of [43]). *If $f$ is a (strongly nondegenerate) self-concordant function, then so is its Fenchel conjugate $f^*$.*

The following result gives a bound on the quadratic approximation of a function, with the distance between two points measured in the local norm. The convergence of Newton's method can be essentially explained by this result.

**Theorem A.8** (Theorem 2.2.2 of [43]). *If $f$ is a self-concordant function, $\mathbf{x} \in dom(f)$, and $\mathbf{y} \in \mathcal{B}_{\mathbf{x}}(\mathbf{x}, 1)$, then*

$$f(\mathbf{y}) \leq f(\mathbf{x}) + \langle \nabla f(\mathbf{x}), \mathbf{y} - \mathbf{x} \rangle + \frac{1}{2} \|\mathbf{y} - \mathbf{x}\|_{\mathbf{x}}^2 + \frac{\|\mathbf{y} - \mathbf{x}\|_{\mathbf{x}}^3}{3(1 - \|\mathbf{y} - \mathbf{x}\|_{\mathbf{x}})},$$

*where $\|\mathbf{y} - \mathbf{x}\|_{\mathbf{x}}^2 := \langle \mathbf{y} - \mathbf{x}, \nabla^2 f(\mathbf{x}) \cdot (\mathbf{y} - \mathbf{x}) \rangle$.*

Finally, we need the following definitions of entropic barrier and universal barrier.

**Definition 2.5** ([10, 14]). Given a convex body $\mathcal{K} \subseteq \mathbb{R}^n$ and some fixed $\theta \in \mathbb{R}^n$, define the function $f(\theta) = \log \left[ \int_{\mathbf{x} \in \mathcal{K}} \exp\langle \mathbf{x}, \theta \rangle d\mathbf{x} \right]$. Then the Fenchel conjugate $f^* : \text{int}(\mathcal{K}) \to \mathbb{R}$ is a self-concordant barrier termed the *entropic barrier*. The entropic barrier is $n$-self-concordant.

**Definition 2.6** ([41, 32]). Given a convex body $\mathcal{K} \subseteq \mathbb{R}^n$, the *universal barrier* of $\mathcal{K}$ is defined as $\psi : \text{int}(\mathcal{K}) \to \mathbb{R}$ by

$$\psi(\mathbf{x}) = \log \text{vol}((\mathcal{K} - \mathbf{x})^\circ).$$

The universal barrier is $n$-self-concordant.

### A.3 Facts from convex geometry

Since our analysis is contingent on the change in the volume of convex bodies when points are added to them or when they are intersected with halfspaces, we invoke the classical result by Grünbaum several times. We therefore state its relevant variants next: Theorem 2.3 applies to log-concave distributions, and Corollary A.10 is its specific case, since the indicator function of a convex set is a log-concave function [8].

**Theorem 2.3** ([20, 11]). *Let $f$ be a log-concave distribution on $\mathbb{R}^d$ with centroid $\mathbf{c}_f$. Let $\mathcal{H} = \left\{ \mathbf{u} \in \mathbb{R}^d : \langle \mathbf{u}, \mathbf{v} \rangle \geq q \right\}$ be a halfspace defined by a normal vector $\mathbf{v} \in \mathbb{R}^d$. Then, $\int_{\mathcal{H}} f(\mathbf{z}) d\mathbf{z} \geq \frac{1}{e} - t^+$, where $t = \frac{q - \langle \mathbf{c}_f, \mathbf{v} \rangle}{\sqrt{\mathbb{E}_{\mathbf{y} \sim f} \langle \mathbf{v}, \mathbf{y} - \mathbf{c}_f \rangle^2}}$ is the distance of the centroid to the halfspace scaled by the standard deviation along the normal vector $\mathbf{v}$ and $t^+ := \max\{0, t\}$.*

*Remark* 9. A crucial special case of Theorem 2.3 is that cutting a convex set through its centroid yields two parts, the smaller of which has volume at least $1/e$ times the original volume and the larger of which is at most $1 - 1/e$ times the original total volume.

**Corollary A.10** ([20]). *Let $\mathcal{K}$ be a convex set with centroid $\mu$ and covariance matrix $\Sigma$. Then, for any point $\mathbf{x}$ satisfying $\|\mathbf{x} - \mu\|_{\Sigma^{-1}} \leq \eta$ and a halfspace $\mathcal{H}$ such that $\mathbf{x} \in \mathcal{H}$, we have $\text{vol}(\mathcal{K} \cap \mathcal{H}) \geq \text{vol}(\mathcal{K}) \cdot (1/e - \eta)$.*

Finally, we need the following facts.

**Fact A.11** (Volumes of standard objects). *The volume of a $q$-dimensional Euclidean ball is given by $\text{vol}(\mathcal{B}_q(0, \bar{R})) = \frac{\pi^{q/2}}{\Gamma(1 + q/2)} \bar{R}^q$, and the volume of a $q$-dimensional cone $= \frac{1}{q+1} \cdot$ volume of base $\cdot$ height.*

## B Our analysis

To analyze Algorithm 1, we define the following potential function that captures the changes in $\mathcal{K}_{\text{in},i}$, $\mathcal{K}_{\text{out},i}$, $t$, and $\mathbf{x}$ in each iteration:

$$\Phi := \underbrace{t\langle \mathbf{c}, \mathbf{x} \rangle + \log \left[ \int_{\widehat{\mathcal{K}}_{\text{out}}} \exp(-t\langle \mathbf{c}, \mathbf{u} \rangle) d\mathbf{u} \right]}_{\text{entropic terms}} + \underbrace{\sum_{i \in [n]} \psi_{\text{in},i}(\mathbf{x}_i)}_{\text{universal terms}}, \tag{B.1}$$

where $\log \left[ \int_{\widehat{\mathcal{K}}_{\text{out}}} \exp(-t\langle \mathbf{c}, \mathbf{u} \rangle) d\mathbf{u} \right]$ is related to the entropic barrier on $\widehat{\mathcal{K}}_{\text{out}}$ (see Appendix B.1) and $\psi_{\text{in},i}$ is the universal barrier on $\mathcal{K}_i$. In the subsequent sections, we study the changes in each of these potential functions along with obtaining bounds on the initial and final potentials and combine them to bound the algorithm's separation oracle complexity.

## B.1 Potential change for the entropic terms

In this section, we study the changes in the entropic terms of Equation (B.1) upon updating the outer convex set $\widehat{\mathcal{K}}_{\text{out}}$ as well as $t$. These two changes are lumped together in this section because both updates affect the term $\log\left[\int_{\widehat{\mathcal{K}}_{\text{out}}} \exp(-t \cdot \langle \mathbf{c}, \mathbf{x}\rangle)\, d\mathbf{x}\right]$, albeit in different ways: the update in $\widehat{\mathcal{K}}_{\text{out}}$ affects it via Grünbaum's Theorem; the update in $t$ affects it via the fact that, by duality with respect to the entropic barrier (Definition 2.5), $\log\left[\int_{\mathbf{x}\in\widehat{\mathcal{K}}_{\text{out}}} \exp(\langle \mathbf{x}, \theta\rangle)d\mathbf{x}\right]$ is also self-concordant. We detail these two potential changes below.

**Lemma B.1** (Potential analysis for outer set). *Let* $\widehat{\mathcal{K}}_{out} := \{\mathbf{x} : \mathbf{x}_i \in \mathcal{K}_{\text{out},i} \cap \{\mathbf{y} : \mathbf{Ay} = \mathbf{b}\}\}$, *and let* $\Phi = \log\left[\int_{\widehat{\mathcal{K}}_{\text{out}}} \exp(-t\langle \mathbf{c}, \mathbf{u}\rangle)d\mathbf{u}\right] + t\langle \mathbf{c}, \mathbf{x}\rangle + \sum_{i\in[n]} \psi_{in,i}(\mathbf{x}_i)$. *Let* $\mathcal{H}_i$ *be the halfspace generated by the separation oracle* $\mathcal{O}_i$ *queried at* $\mathbf{x}^\star_{out,i}$ *as shown in Line 17 of Algorithm 1. Then the new potential* $\Phi^{(new)} = \log\left[\int_{\widehat{\mathcal{K}}_{\text{out}}\cap\mathcal{H}_i} \exp(-t\langle \mathbf{c}, \mathbf{u}\rangle)d\mathbf{u}\right] + t\langle \mathbf{c}, \mathbf{x}\rangle + \sum_{i\in[n]} \psi_{in,i}(\mathbf{x}_i)$ *is bounded from above as follows.*

$$\Phi^{(new)} \leq \Phi + \log(1 - 1/e).$$

*Proof.* The change in potential is given by

$$\Phi^{(\text{new})} - \Phi = \log\left[\frac{\int_{\widehat{\mathcal{K}}_{\text{out}}\cap\mathcal{H}_i} \exp(-t \cdot \langle \mathbf{c}, \mathbf{x}\rangle)\, d\mathbf{x}}{\int_{\widehat{\mathcal{K}}_{\text{out}}} \exp(-t \cdot \langle \mathbf{c}, \mathbf{x}\rangle)\, d\mathbf{x}}\right].$$

We now apply Theorem 2.3 to the right hand side, with the function $f(\mathbf{x}) = \exp(-t \cdot \langle \mathbf{c}, \mathbf{x}\rangle - A(t\mathbf{c}))$, where $A(\theta) = \log\left[\int_{\widehat{\mathcal{K}}_{\text{out}}} \exp(-\langle \theta, \mathbf{x}\rangle)d\mathbf{x}\right]$. Noting that each halfspace $\mathcal{H}_i$ passes directly through $\mathbf{x}^\star_{\text{out},i}$, where $\mathbf{x}^\star_{\text{out}}$ is the centroid of $\widehat{\mathcal{K}}_{\text{out}}$ with respect to $f$ (by the definition of $\mathbf{x}^\star_{\text{out}}$ in Equation (3.5)), Remark 9 applies and gives the claimed volume change. $\square$

To capture the change in potential due to the update in $t$, we recall the alternative perspective to the function $\log\left[\int_{\widehat{\mathcal{K}}_{\text{out}}} \exp(-t\langle \mathbf{c}, \mathbf{x}\rangle)d\mathbf{x}\right]$ given by Definition 2.5 and derive properties of self-concordant barriers.

**Lemma B.2.** *Consider a* $\nu$-*self-concordant barrier* $\psi : int(\mathcal{K}) \to \mathbb{R}$ *over the interior of a convex set* $\mathcal{K} \subseteq \mathbb{R}^d$. *Define*

$$\xi_t^\psi := \min_{\mathbf{x}}\left[t \cdot \langle \mathbf{c}, \mathbf{x}\rangle + \psi(\mathbf{x})\right] \text{ and } \mathbf{x}_t := \arg\min_{\mathbf{x}}\left[t \cdot \langle \mathbf{c}, \mathbf{x}\rangle + \psi(\mathbf{x})\right]. \tag{B.2}$$

*Then for* $0 \leq h \leq \frac{1}{\sqrt{\nu}}$, *we have*

$$\xi_t^\psi + th \cdot \langle \mathbf{x}_t, \mathbf{c}\rangle \geq \xi_{t(1+h)}^\psi \geq \xi_t^\psi + ht \cdot \langle \mathbf{c}, \mathbf{x}_t\rangle - h^2\nu.$$

*Proof.* Note that here the first inequality is fairly generic and holds for any function $\psi$. By definition of $\xi_{t(1+h)}^\psi$ and $\xi_t^\psi$ in Equation (B.2) and using the fact that the value on the right hand side of Equation (B.3) is smaller than the expression evaluated at a fixed $\mathbf{x} = \mathbf{x}_t$, we have

$$\begin{aligned}
\xi_{t(1+h)}^\psi &= \min_{\mathbf{x}}\left[t(1+h) \cdot \langle \mathbf{x}, \mathbf{c}\rangle + \psi(\mathbf{x})\right] \tag{B.3}\\
&\leq t(1+h) \cdot \langle \mathbf{x}_t, \mathbf{c}\rangle + \psi(\mathbf{x}_t)\\
&= \xi_t^\psi + th \cdot \langle \mathbf{x}_t, \mathbf{c}\rangle.
\end{aligned}$$

We now prove the second inequality of the lemma. This one specifically uses the self-concordance of $\psi$. Observe first, by definition,

$$\xi_t^\psi = -\psi^*(-t\mathbf{c}). \tag{B.4}$$

Since $\psi$ is a self-concordant barrier (and hence, a self-concordant function), Theorem A.7 implies that $\psi^*$ is a self-concordant function as well. Then, by applying Theorem A.8 to $\psi^*$ under the assumption $\|-th\mathbf{c}\|_{-t\mathbf{c}} \leq 1$ yields

$$\psi^*(-t\mathbf{c} - th\mathbf{c}) \leq \psi^*(-t\mathbf{c}) + \langle \nabla\psi^*(-t\mathbf{c}), -th\mathbf{c}\rangle + \left[\frac{1}{2}\|-th\mathbf{c}\|_{-t\mathbf{c}}^2 + \frac{\|-th\mathbf{c}\|_{-t\mathbf{c}}^3}{3(1 - \|-th\mathbf{c}\|_{-t\mathbf{c}})}\right]. \tag{B.5}$$

By applying the first-order optimality condition to the definition of $\mathbf{x}_t$ in Equation (B.2), we see that
$$\nabla \psi(\mathbf{x}_t) = -t\mathbf{c}. \tag{B.6}$$
Next, evaluating $a := \| - th\mathbf{c}\|_{-t\mathbf{c}}$ to check the assumption $\| - th\mathbf{c}\|_{-t\mathbf{c}} \leq 1$, we get
$$\begin{aligned} a^2 = h^2 \langle -t\mathbf{c}, \nabla^2 \psi^*(-t\mathbf{c}) \cdot (-t\mathbf{c}) \rangle &= h^2 \langle \nabla \psi(\mathbf{x}_t), \nabla^2 \psi^*(\nabla \psi(\mathbf{x}_t)) \cdot \nabla \psi(\mathbf{x}_t) \rangle \\ &= h^2 \langle \nabla \psi(\mathbf{x}_t), (\nabla^2 \psi(\mathbf{x}_t))^{-1} \cdot \nabla \psi(\mathbf{x}_t) \rangle \\ &\leq h^2 \nu \end{aligned}$$
where we used Equation (B.6) and Lemma A.4, in the first two equations and Definition 2.4 and the complexity value of $\psi$ in the last step. Our range of $h$ proves that $a \leq 1$, which is what we need for Inequality (B.5) to hold. We continue our computation to get
$$\left[ \frac{1}{2} \| - th\mathbf{c}\|_{-t\mathbf{c}}^2 + \frac{\| - th\mathbf{c}\|_{-t\mathbf{c}}^3}{3(1 - \| - th\mathbf{c}\|_{-t\mathbf{c}})} \right] \leq \frac{1}{2} h^2 \nu + \frac{1}{3} h^3 \nu^{3/2} \leq \frac{1}{2} h^2 \nu + \frac{1}{3} h^2 \nu \leq h^2 \nu. \tag{B.7}$$
Applying Lemma A.3 to Equation (B.6) gives
$$\nabla \psi^*(-t\mathbf{c}) = \mathbf{x}_t. \tag{B.8}$$
Plugging Equation (B.8) and Inequality (B.7) into the first and second-order terms, respectively, of Inequality (B.5) gives
$$\psi^*(-t\mathbf{c} - th\mathbf{c}) \leq \psi^*(-t\mathbf{c}) + \langle \mathbf{x}_t, -th\mathbf{c} \rangle + h^2 \nu.$$
Plugging in Equation (B.4) gives the desired inequality and completes the proof. $\qquad\square$

To finally compute the potential change due to $t$, we need the following result about the self-concordance parameter of the entropic barrier. While [10] prove that this barrier on a set in $\mathbb{R}^d$ is $(1 + \epsilon_d)d$-self-concordant, the recent work of [14] remarkably improves this complexity to exactly $d$.

**Theorem B.3** ([14]). *The entropic barrier on any convex body $\mathcal{K} \subseteq \mathbb{R}^d$ is a $d$-self-concordant barrier.*

We may now compute the potential change due to change in $t$ in Line 9.

**Lemma B.4.** *When $t$ is updated to $t \cdot \left[1 + \frac{\eta}{4m}\right]$ in Line 9 of Algorithm 1, the potential $\Phi$ Equation (B.1) increases to $\Phi^{(new)}$ as follows:*
$$\Phi^{(new)} \leq \Phi + \eta + \eta^2.$$

*Proof.* Recall that the barrier function we use for the set $\widehat{\mathcal{K}}_{\text{out}}$ is the entropic barrier $\psi_{\text{out}}$. By Equation (B.2) and the definition of conjugate, we have
$$-\xi_t^{\psi_{\text{out}}} = \max_{\mathbf{v}} \left[ \langle -t\mathbf{c}, \mathbf{v} \rangle - \psi_{\text{out}}(\mathbf{v}) \right] = \psi_{\text{out}}^*(-t\mathbf{c}).$$
Applying Definition 2.5, taking the conjugate on both sides of the preceding equation, and using Lemma A.2 then gives
$$-\xi_t^{\psi_{\text{out}}} = \log \left[ \int_{\widehat{\mathcal{K}}_{\text{out}}} \exp(-t \cdot \langle \mathbf{c}, \mathbf{u} \rangle) \, d\mathbf{u} \right]. \tag{B.9}$$
From Equation (B.1), the change in potential by changing $t$ to $t \cdot (1 + h)$ for some $h > 0$ may be expressed as
$$\Phi^{(new)} - \Phi = \log \left[ \int_{\widehat{\mathcal{K}}_{\text{out}}} \exp\langle -t(1+h)\mathbf{c}, \mathbf{v} \rangle d\mathbf{v} \right] - \log \left[ \int_{\widehat{\mathcal{K}}_{\text{out}}} \exp\langle -t\mathbf{c}, \mathbf{v} \rangle d\mathbf{v} \right] + \langle th \cdot \mathbf{c}, \mathbf{x} \rangle.$$
By applying $h = \frac{\eta}{4m}$ and $\nu = m$ (via a direct application of Theorem B.3), we have $h = \frac{\eta}{4m} \leq \frac{1}{\sqrt{m}} = \frac{1}{\sqrt{\nu}}$, and so we may now apply Equation (B.9) and Lemma B.2 in the preceding equation to obtain the following bound.
$$\Phi^{(new)} - \Phi \leq th\langle \mathbf{c}, \mathbf{x} \rangle - th\langle \mathbf{c}, \mathbf{x}_t \rangle + h^2 \nu.$$
From Equation (3.2) and Equation (B.2), we see that $\mathbf{x}_t$ for the entropic barrier satisfies the equation $\mathbf{x}_t = \mathbf{x}_{\text{out}}^\star$, and applying the guarantee $\langle \mathbf{c}, \mathbf{x} \rangle \leq \langle \mathbf{c}, \mathbf{x}_{\text{out}}^\star \rangle + \frac{4m}{t}$ to this inequality, we obtain
$$\Phi^{(new)} - \Phi \leq th \cdot \frac{4m}{t} + h^2 \nu = \eta + \left( \frac{\eta}{4m} \right)^2 \nu \leq \eta + \eta^2.$$
$\qquad\square$

## B.2 Potential change for the universal terms

In this section, we study the change in volume on growing the inner convex set $\mathcal{K}_{\text{in},i}$ in Line 15. As mentioned in Section 3, our barrier of choice on this set is the universal barrier introduced in [38] (see also [21]). This barrier was constructed to demonstrate that *any* convex body in $\mathbb{R}^n$ admits an $O(n)$-self-concordant barrier, and its complexity parameter was improved to exactly $n$ in [32].

Conceptually, we choose the universal barrier for the inner set because the operation we perform on the inner set (i.e., generating its convex hull with an external point $\mathbf{x}_{\text{out}}^\star$) is dual to the operation of intersecting the outer set with the separating halfspace containing $\mathbf{x}_{\text{out}}^\star$ (see Lemma 2.2), which suggests the use of a barrier dual to the entropic barrier used on the outer set. As explained in [10], for the special case of convex cones, the universal barrier is precisely one such barrier.

We now state a technical property of the universal barrier, which we use in the potential argument for this section.

**Lemma B.5** ([32, Lemma 1], [38, 21]). *Given a convex set $\mathcal{K} \in \mathbb{R}^d$ and $\mathbf{x} \in \mathcal{K}$, let $\psi_\mathcal{K}(\mathbf{x}) := \log \text{vol}(\mathcal{K} - \mathbf{x})^\circ$ be the universal barrier defined on $\mathcal{K}$ with respect to $\mathbf{x}$. Let $\mu \in \mathbb{R}^d$ be the center of gravity and $\Sigma \in \mathbb{R}^{d \times d}$ be the covariance matrix of the body $(\mathcal{K} - \mathbf{x})^\circ$, where $(\mathcal{K} - \mathbf{x})^\circ = \{\mathbf{y} \in \mathbb{R}^n : \mathbf{y}^\top(\mathbf{z} - \mathbf{x}) \leq 1, \forall \mathbf{z} \in \mathcal{K}\}$ is the polar set of $\mathcal{K}$ with respect to $\mathbf{x}$. Then, we have that*

$$\nabla \psi_\mathcal{K}(\mathbf{x}) = (d+1)\mu, \ \nabla^2 \psi_\mathcal{K}(\mathbf{x}) = (d+1)(d+2)\Sigma + (d+1)\mu\mu^\top.$$

**Lemma B.6.** *Given a convex set $\mathcal{K} \subseteq \mathbb{R}^d$ and a point $\mathbf{x} \in \mathcal{K}$. Let $\psi_\mathcal{K} := \log \text{vol}(\mathcal{K} - \mathbf{x})^\circ$ be the universal barrier defined on $\mathcal{K}$ with respect to $\mathbf{x}$. Let $\eta \leq 1/4$ and $\mathbf{y} \in \mathcal{K}$ be a point satisfying the following condition*

$$\langle \nabla \psi_\mathcal{K}(\mathbf{x}), \mathbf{y} - \mathbf{x} \rangle + \eta \|\mathbf{y} - \mathbf{x}\|_\mathbf{x} \geq 4d, \tag{B.10}$$

*and construct the new set $\text{conv}\{\mathcal{K}, \mathbf{y}\}$. Then, the value of the universal barrier defined on this new set with respect to $\mathbf{x}$ satisfies the following inequality.*

$$\psi_{\mathcal{K},new}(\mathbf{x}) := \psi_{\text{conv}\{\mathcal{K},\mathbf{y}\}}(\mathbf{x}) = \log \text{vol}(\text{conv}(\mathcal{K}, \mathbf{y}) - \mathbf{x})^\circ \leq \psi_\mathcal{K}(\mathbf{x}) + \log(1 - 1/e + \eta).$$

*Proof.* By Lemma 2.2, we have that

$$(\text{conv}(\mathcal{K}, \mathbf{y}) - \mathbf{x})^\circ \subseteq (\mathcal{K} - \mathbf{x})^\circ \cap \mathcal{H},$$

where $\mathcal{H} = \{\mathbf{z} \in \mathbb{R}^n : \langle \mathbf{z}, \mathbf{y} - \mathbf{x} \rangle \leq 1\}$. Our strategy to computing the deviation of $\psi_{\mathcal{K},\text{new}}(\mathbf{x}) := \psi_{\text{conv}(\mathcal{K},\mathbf{y})}(\mathbf{x}) = \log \text{vol}(\text{conv}(\mathcal{K}, \mathbf{y}) - \mathbf{x})^\circ$ from $\psi_\mathcal{K}(\mathbf{x})$ is to compute the change in $\text{vol}(\text{conv}(\mathcal{K}, \mathbf{y}) - \mathbf{x})^\circ \leq \text{vol}[(\mathcal{K} - \mathbf{x})^\circ \cap \mathcal{H}]$ from $\text{vol}(\mathcal{K} - \mathbf{x})^\circ$, for which it is immediate that one may apply an appropriate form of Grünbaum's Theorem.

Let $\mu$ be the center of gravity of the body $(\mathcal{K} - \mathbf{x})^\circ$. If $\mu \notin \mathcal{H}$, then Corollary A.10 (with $\eta = 0$) gives

$$\text{vol}[(\mathcal{K} - \mathbf{x})^\circ \cap \mathcal{H}] \leq \text{vol}(\mathcal{K} - \mathbf{x})^\circ \cdot (1 - 1/e),$$

and taking the logarithm on both sides gives the claimed bound. We now consider the case in which $\mu \in \mathcal{H}$, and the variance matrix of the body $(\mathcal{K} - \mathbf{x})^\circ$ is $\Sigma$. Define $\mathbf{v} = \mathbf{y} - \mathbf{x}$, and consider the point

$$\mathbf{z} = \mu + \frac{1 - \langle \mathbf{v}, \mu \rangle}{\|\mathbf{v}\|_\Sigma^2} \cdot \Sigma \mathbf{v}.$$

This point satisfies $\langle \mathbf{v}, \mathbf{z} \rangle = 1$, which implies $\mathbf{z} \in \mathcal{H}$. Specifically, $\mathbf{z}$ lies on the separating hyperplane. We show that $\mathbf{z}$ is sufficiently close to $\mu$, so that even though $\mu \in \mathcal{H}$, the subset of $(\mathcal{K} - \mathbf{x})^\circ$ cut out by the halfspace $\mathcal{H}$ is not too large. By applying Lemma B.5 to compute $\|\mathbf{v}\|_\mathbf{x}^2 = (d+1)(d+2)\|\mathbf{v}\|_\Sigma^2 + (d+1)\langle \mathbf{v}, \mu \rangle^2$, we may compute the following quantity.

$$\begin{aligned}
\|\mathbf{z} - \mu\|_{\Sigma^{-1}} &= \frac{1 - \langle \mathbf{v}, \mu \rangle}{\sqrt{\frac{1}{(d+1)(d+2)}\|\mathbf{v}\|_\mathbf{x}^2 - \frac{1}{d+2} \cdot \langle \mathbf{v}, \mu \rangle^2}} \\
&= \sqrt{(d+1)(d+2)} \cdot \frac{1 - \langle \mathbf{v}, \mu \rangle}{\sqrt{\frac{1}{2}\|\mathbf{v}\|_\mathbf{x}^2 + \frac{1}{2}\|\mathbf{v}\|_\mathbf{x}^2 - (d+1)\langle \mathbf{v}, \mu \rangle^2}}.
\end{aligned} \tag{B.11}$$

Applying the expression for gradient from Lemma B.5 in Equation (B.10), we have

$$\eta \|\mathbf{v}\|_\mathbf{x} \geq 4d - (d+1)\langle \mathbf{v}, \mu \rangle \geq 2d\langle \mathbf{v}, \mu \rangle,$$

where we used the fact that $\mu \in \mathcal{H}$ implies $\langle \mathbf{v}, \mu \rangle \leq 1$. Since $\eta \leq 1/4$, we have $\frac{1}{2}\|\mathbf{v}\|_{\mathbf{x}}^2 \geq (d+1)\langle \mathbf{v}, \mu \rangle^2$. Plugging this in Equation (B.11) gives

$$\|\mathbf{z} - \mu\|_{\mathbf{A}^{-1}} \leq \sqrt{(d+1)(d+2)} \cdot \frac{1 - \langle \mathbf{v}, \mu \rangle}{\sqrt{\frac{1}{2}\|\mathbf{v}\|_{\mathbf{x}}^2}} \leq 4d \frac{1 - \langle \mathbf{v}, \mu \rangle}{\|\mathbf{v}\|_{\mathbf{x}}}$$

$$\leq 4d \cdot \frac{1 - \langle \mathbf{v}, \mu \rangle}{4d(1 - \langle \mathbf{v}, \mu \rangle)/\eta} \leq \eta,$$

which implies Corollary A.10 applies, giving us the desired volume reduction. $\square$

## B.3  Potential change for the update of $\mathbf{x}$

In this section, we quantify the amount of progress made in Line 22 of Algorithm 1 by computing the change in the potential $\Phi$ as defined in Equation (B.1).

**Lemma B.7.** *Consider the potential $\Phi$ Equation (B.1) and the update step $\delta_{\mathbf{x}} = \frac{\eta}{2} \cdot \frac{\mathbf{x}_{out}^\star - \mathbf{x}}{\|\mathbf{x}_{out}^\star - \mathbf{x}\|_{\mathbf{x},1}}$ as in Line 22. Assume the guarantees in Inequality (3.3) and Inequality (3.4). Then the potential $\Phi$ incurs the following minimum decrease.*

$$\Phi^{(new)} \leq \Phi - \frac{\eta^2}{4}.$$

*Proof.* Taking the gradient of $\Phi$ with respect to $\mathbf{x}$ and rearranging the terms gives

$$t\mathbf{c} = \nabla_{\mathbf{x}}\Phi - \sum_{i=1}^{n} \nabla\psi_{\text{in},i}(\mathbf{x}_i). \tag{B.12}$$

By applying the expression for $t\mathbf{c}$ from the preceding equation, we get

$$\Phi^{(new)} - \Phi = t\langle \mathbf{c}, \mathbf{x} + \delta_{\mathbf{x}} \rangle + \sum_{i=1}^{n} \psi_{\text{in},i}(\mathbf{x}_i + \delta_{\mathbf{x},i}) - t\langle \mathbf{c}, \mathbf{x} \rangle - \sum_{i=1}^{n} \psi_{\text{in},i}(\mathbf{x}_i)$$

$$= \langle \nabla_{\mathbf{x}}\Phi, \delta_{\mathbf{x}} \rangle + \sum_{i=1}^{n} \underbrace{[\psi_{\text{in},i}(\mathbf{x}_i + \delta_{\mathbf{x},i}) - \psi_{\text{in},i}(\mathbf{x}_i) - \langle \nabla\psi_{\text{in},i}(\mathbf{x}_i), \delta_{\mathbf{x},i} \rangle]}_{q_{\psi_{\text{in},i}}(\mathbf{x}_i)}. \tag{B.13}$$

The term $q_{\psi_{\text{in},i}}(\mathbf{x}_i)$ measures the error due to first-order approximation of $\psi_{\text{in},i}$ around $\mathbf{x}_i$. Since $\psi_{\text{in},i}(\mathbf{x}_i)$ is self-concordant functions and $\|\delta_{\mathbf{x},i}\|_{\mathbf{x}_i} \leq \|\delta_{\mathbf{x}}\|_{\mathbf{x},1} \leq \eta \leq 1/4$, Theorem A.8 shows that

$$\psi_{\text{in},i}(\mathbf{x}_i + \delta_{\mathbf{x},i}) - \psi_{\text{in},i}(\mathbf{x}_i) - \langle \nabla\psi_{\text{in},i}(\mathbf{x}_i), \delta_{\mathbf{x},i} \rangle \leq \|\delta_{\mathbf{x},i}\|_{\mathbf{x},i}^2. \tag{B.14}$$

Plugging in Inequality (B.14) into Equation (B.13), we get

$$\Phi^{(new)} - \Phi \leq \langle \nabla_{\mathbf{x}}\Phi, \delta_{\mathbf{x}} \rangle + \|\delta_{\mathbf{x}}\|_{\mathbf{x},1}^2. \tag{B.15}$$

We now bound the two terms on the right hand side one at a time. Using the definition of $\delta_{\mathbf{x}}$ (as given in the statement of the lemma) and of $\nabla_{\mathbf{x}}\Phi$ from Equation (B.12) gives

$$\langle \nabla_{\mathbf{x}}\Phi, \delta_{\mathbf{x}} \rangle = \frac{\eta}{2} \frac{1}{\|\mathbf{x}_{out}^\star - \mathbf{x}\|_{\mathbf{x},1}} \langle \nabla_{\mathbf{x}}\Phi, \mathbf{x}_{out}^\star - \mathbf{x} \rangle$$

$$= \frac{\eta}{2} \frac{1}{\|\mathbf{x}_{out}^\star - \mathbf{x}\|_{\mathbf{x},1}} \left[ \langle t\mathbf{c}, \mathbf{x}_{out}^\star - \mathbf{x} \rangle + \sum_{i=1}^{n} \langle \nabla\psi_{\text{in},i}(\mathbf{x}_i), \mathbf{x}_{out,i}^\star - \mathbf{x}_i \rangle \right]$$

$$\leq \frac{\eta}{2} \frac{1}{\|\mathbf{x}_{out}^\star - \mathbf{x}\|_{\mathbf{x},1}} \left[ \langle t\mathbf{c}, \mathbf{x}_{out}^\star - \mathbf{x} \rangle + \sum_{i=1}^{n} \left( 4r_i - \eta\|\mathbf{x}_{out,i}^\star - \mathbf{x}_i\|_{\mathbf{x}_i} \right) \right]$$

$$= \frac{\eta}{2} \frac{1}{\|\mathbf{x}_{out}^\star - \mathbf{x}\|_{\mathbf{x},1}} \left[ \langle t\mathbf{c}, \mathbf{x}_{out}^\star - \mathbf{x} \rangle + 4m - \eta\|\mathbf{x}_{out}^\star - \mathbf{x}\|_{\mathbf{x},1} \right]$$

$$\leq \frac{\eta}{2} \frac{1}{\|\mathbf{x}_{out}^\star - \mathbf{x}\|_{\mathbf{x},1}} \cdot (-\eta\|\mathbf{x}_{out}^\star - \mathbf{x}\|_{\mathbf{x},1})$$

$$= -\eta^2/2. \tag{B.16}$$

where the third step follows from Inequality (3.3), the fourth step follows from $\sum_{i=1}^{n} d_i = m$, and the fifth step follows from Inequality (3.4). To bound the second term, we note from Line 22 that

$$\|\delta_{\mathbf{x}}\|_{\mathbf{x},1}^2 = \left( \frac{\eta}{2} \cdot \frac{\|\mathbf{x}_{\text{out}}^\star - \mathbf{x}\|_{\mathbf{x},1}}{\|\mathbf{x}_{\text{out}}^\star - \mathbf{x}\|_{\mathbf{x},1}} \right)^2 = \eta^2/4. \tag{B.17}$$

Hence, we may plug in Inequality (B.16) and Equation (B.17) into Inequality (B.15) to get the desired result. □

## B.4 Total oracle complexity

Before we bound the total oracle complexity of the algorithm, we first bound the total potential change throughout the algorithm.

**Lemma B.8.** *Consider the potential function* $\Phi = t\langle \mathbf{c}, \mathbf{x} \rangle + \log \left[ \int_{\widehat{\mathcal{K}}_{out}} \exp(-t\langle \mathbf{c}, \mathbf{u} \rangle) d\mathbf{u} \right] + \sum_{i \in [n]} \psi_{in,i}(\mathbf{x}_i)$ *as defined in Equation* (B.1) *associated with Algorithm 1. Let* $\Phi_{init}$ *be the potential at* $t = t_{init}$ *of this algorithm, and let* $\Phi_{end}$ *be the potential at* $t = t_{end}$*. Suppose at* $t = t_{init}$ *in Algorithm 1, we have* $\mathcal{B}_m(\mathbf{x}, \bar{r}) \subseteq \mathcal{K}_{in}$ *with* $\bar{r} = r / \text{poly}(m)$ *and* $\mathcal{K}_{out} \subseteq \mathcal{B}_m(0, \bar{R})$ *with* $\bar{R} = O(\sqrt{n}R)$*. Then we have, under the assumptions of Theorem B.10, that*

$$\Phi_{init} - \Phi_{end} \leq O\left( m \log \left( \frac{mR}{\epsilon r} \right) \right).$$

*Proof.* For this proof, we introduce the following notation: let $\text{vol}_{\mathbf{A}}(\cdot)$ denote the volume restricted to the subspace $\{\mathbf{x} : \mathbf{A}\mathbf{x} = \mathbf{b}\}$. We also invoke Fact A.11. We now bound the change in the potential term by term, starting with the entropic terms

$$t\langle \mathbf{c}, \mathbf{x} \rangle + \log \left[ \int_{\widehat{\mathcal{K}}_{out}} \exp(-t\langle \mathbf{c}, \mathbf{u} \rangle) d\mathbf{u} \right] \tag{B.18}$$

at $t = t_{\text{init}}$ and a lower bound on it at $t = t_{\text{end}}$. We start with bounding Equation (B.18) evaluated at $t = t_{\text{end}} = \frac{8m}{\epsilon \|\mathbf{c}\|_2 R}$.

Let $\bar{\mathbf{x}} = \arg \min_{\mathbf{x} \in \widehat{\mathcal{K}}_{out}} \langle \mathbf{c}, \mathbf{x} \rangle$ and $\alpha = \langle \mathbf{c}, \bar{\mathbf{x}} \rangle$. By optimality of $\bar{\mathbf{x}}$, we know that $\bar{\mathbf{x}} \in \partial \widehat{\mathcal{K}}_{out}$. Denote $\mathcal{B}_A(\mathbf{z}, \bar{r})$ to be $\mathcal{B}(\mathbf{z}, \bar{r})$ restricted to the subspace $\{\mathbf{x} : \mathbf{A}\mathbf{x} = \mathbf{b}\}$. Note that $\widehat{\mathcal{K}}_{out} \supseteq \mathcal{B}_{\mathbf{A}}(\mathbf{z}, \bar{r})$. Consider the cone $\mathcal{C}$ and halfspace $\mathcal{H}$ defined by

$$\mathcal{C} = \bar{\mathbf{x}} + \{\lambda \mathbf{y} : \lambda > 0, \mathbf{y} \in \mathcal{B}_{\mathbf{A}}(\mathbf{z} - \bar{\mathbf{x}}, \bar{r})\} \text{ and } \mathcal{H} := \left\{ \mathbf{x} : \langle \mathbf{c}, \mathbf{x} \rangle \leq \alpha + \frac{1}{t_{\text{end}}} \right\}.$$

Then, by a similarity argument, we note that $\mathcal{C} \cap \mathcal{H}$ contains a cone with height $\frac{1}{t_{\text{end}}\|c\|_2}$ and base radius $\frac{\bar{r}}{\bar{R} t_{\text{end}}\|c\|_2}$, which means

$$\text{vol}_{\mathbf{A}}(\mathcal{C} \cap \mathcal{H}) \geq \frac{1}{m - \text{rank}(\mathbf{A})} \cdot \frac{1}{t_{\text{end}}\|c\|_2} \cdot \left( \frac{\bar{r}}{\bar{R} t_{\text{end}}\|c\|_2} \right)^{m - \text{rank}(\mathbf{A}) - 1} \cdot \text{vol}(\mathcal{B}_{m - \text{rank}(\mathbf{A}) - 1}(0, 1)).$$

Then, we have

$$\log\left[\int_{\widehat{\mathcal{K}}_{\text{out}}}\exp(-t_{\text{end}}\langle\mathbf{c},\mathbf{u}\rangle)d\mathbf{u}\right]+t_{\text{end}}\langle\mathbf{c},\mathbf{x}\rangle \geq \log\left[\int_{\widehat{\mathcal{K}}_{\text{out}}}\exp(-t_{\text{end}}\langle\mathbf{c},\mathbf{u}\rangle)d\mathbf{u}\right]+t_{\text{end}}\min_{\mathbf{x}\in\widehat{\mathcal{K}}_{\text{out}}}\langle\mathbf{c},\mathbf{x}\rangle$$

$$\geq \log\left[\int_{\mathcal{C}\cap\mathcal{H}}\exp(-t_{\text{end}}\langle\mathbf{c},\mathbf{u}\rangle)d\mathbf{u}\right]+t_{\text{end}}\alpha$$

$$\geq \log\left[\int_{\mathcal{C}\cap\mathcal{H}}\exp(-t_{\text{end}}\alpha-1)d\mathbf{u}\right]+t_{\text{end}}\alpha$$

$$= \log\left[\frac{1}{e}\cdot\text{vol}_{\mathbf{A}}(\mathcal{C}\cap\mathcal{H})\exp(-t_{\text{end}}\alpha)\right]+t_{\text{end}}\alpha$$

$$= \log\left[\text{vol}_{\mathbf{A}}(\mathcal{C}\cap\mathcal{H})\cdot\frac{1}{e}\right]$$

$$\geq -(m-\text{rank}(\mathbf{A})-1)\cdot\log(\bar{R}t_{\text{end}}\|c\|_2/\bar{r})$$
$$+\log(\text{vol}(\mathcal{B}_{m-\text{rank}(\mathbf{A})-1}(0,1)))$$
$$-\log(m-\text{rank}(\mathbf{A}))-\log(t_{\text{end}}\|\mathbf{c}\|_2)-1. \tag{B.19}$$

Next, to bound Equation (B.18) at $t=t_{\text{init}}$, we may express these terms as follows.

$$\log\left[\int_{\widehat{\mathcal{K}}_{\text{out}}}\exp(-t_{\text{init}}\cdot\langle\mathbf{c},\mathbf{u}\rangle)d\mathbf{u}\right]+t_{\text{init}}\cdot\langle\mathbf{c},\mathbf{x}\rangle \leq \log\left[\text{vol}_{\mathbf{A}}(\widehat{\mathcal{K}}_{\text{out}})\right]+t_{\text{init}}\cdot\max_{\mathbf{u}\in\widehat{\mathcal{K}}_{\text{out}}}\langle\mathbf{c},\mathbf{x}-\mathbf{u}\rangle$$

$$\leq \log(\text{vol}(\mathcal{B}_{m-\text{rank}(\mathbf{A})}(0,\bar{R})))+t_{\text{init}}\cdot 2\bar{R}\|\mathbf{c}\|_2$$

$$\leq \log(\text{vol}(\mathcal{B}_{m-\text{rank}(\mathbf{A})}(0,1)))$$
$$+(m-\text{rank}(\mathbf{A}))\log\bar{R}+O(m\log m), \tag{B.20}$$

where the second step is by $\widehat{\mathcal{K}}_{\text{out}}\subseteq\mathcal{K}_{\text{out}}\subseteq\mathcal{B}_{\sum_{i\in[n]}d_i}(0,\bar{R})$ (here, the second inclusion is by assumption), and the third step is by $\text{vol}(\mathcal{B}_q(0,\bar{R}))=\frac{\pi^{q/2}}{\Gamma(1+q/2)}\bar{R}^q$ and our choice of $t_{\text{init}}:=\frac{m\log m}{\sqrt{n}\|\mathbf{c}\|_2 R}$.

We now compute the change in the entropic barrier $\sum_{i\in[n]}\psi_{\text{in},i}(\mathbf{x}_i)$, where

$$\psi_{\text{in},i}(\mathbf{x}_i)=\log\text{vol}(\mathcal{K}_{\text{in},i}^{\circ}(\mathbf{x}_i)).$$

Define $\mathcal{B}_d(0,r)$ to be the $d$-dimensional Euclidean ball centred at the origin and with radius $r$. We note by the radius assumption of Theorem B.10 that $\mathcal{K}_{\text{in},i}\subseteq\mathcal{K}_i\subseteq\mathcal{B}_{d_i}(0,\bar{R})$ throughout the algorithm. By the assumption made in this lemma's statement, we have that at the start of Algorithm 1, $\mathcal{K}_{\text{in},i}\supseteq\mathcal{B}_{d_i}(\mathbf{x},\bar{r})$. These give us the following bounds.

$$\psi_{\text{in},i}^{\text{end}}(\mathbf{x}_i)\geq\log(\text{vol}(\mathcal{B}_{d_i}^{\circ}(0,\bar{R}))\quad\text{and}\quad\psi_{\text{in},i}^{\text{init}}(\mathbf{x}_i)\leq\log(\text{vol}(\mathcal{B}_{d_i}^{\circ}(\mathbf{x}_i,\bar{r}))).$$

Applying the fact that $\text{vol}(\mathcal{B}_d(0,r))\propto r^d$ and summing over all $i\in[n]$ gives

$$\sum_{i\in[n]}\left[\psi_{\text{in},i}^{\text{init}}(\mathbf{x}_i)-\psi_{\text{in},i}^{\text{end}}(\mathbf{x}_i)\right]\leq\sum_{i\in[n]}\log\left(\frac{\text{vol}(\mathcal{B}_{d_i}(\mathbf{x}_i,1/\bar{r}))}{\text{vol}(\mathcal{B}_{d_i}(0,1/\bar{R}))}\right)$$

$$=\sum_{i\in[n]}d_i\log(\bar{R}/\bar{r})=m\log(\bar{R}/\bar{r}). \tag{B.21}$$

Combining Inequality (B.20), Inequality (B.19), and Inequality (B.21), we have
$$\Phi_{\text{init}}-\Phi_{\text{end}}\leq m\log(mR/r)$$
$$+\left[\log(\text{vol}(\mathcal{B}_{m-\text{rank}(\mathbf{A})}(0,1)))+(m-\text{rank}(\mathbf{A}))\log\bar{R}+O(m\log m)\right]$$
$$+(m-\text{rank}(\mathbf{A})-1)\cdot\log(\bar{R}t_{\text{end}}\|c\|_2/\bar{r})-\log(\text{vol}(\mathcal{B}_{m-\text{rank}(\mathbf{A})-1}(0,1)))$$
$$+\log(m-\text{rank}(\mathbf{A}))+\log(t_{\text{end}}\|\mathbf{c}\|_2)+1$$
$$\leq m\log(mR/\epsilon r)$$
$$+O(m\log m)$$
$$+O((m-\text{rank}(\mathbf{A}))\log(mR/\epsilon r))\leq O(m\log(mR/\epsilon r)).$$

$$\square$$

**Lemma B.9.** *[Total oracle complexity] Suppose the inputs $\mathcal{K}_{in}$ and $\mathcal{K}_{out}$ to Algorithm 1 satisfy $\mathcal{K}_{out} \subseteq \mathcal{B}_m(0, \bar{R})$ with $\bar{R} = O(\sqrt{n}R)$ and $\mathcal{K}_{in} \supseteq \mathcal{B}(\mathbf{z}, \bar{r})$ with $\bar{r} = r/\operatorname{poly}(m)$. Then, when Algorithm 1 terminates at $t \geq t_{end}$, it outputs a solution $\mathbf{x}$ that satisfies*

$$\mathbf{c}^\top \mathbf{x} \leq \min_{\mathbf{x} \in \mathcal{K}, A\mathbf{x} = \mathbf{b}} \mathbf{c}^\top \mathbf{x} + \epsilon \cdot \|\mathbf{c}\|_2 R$$

*using at most $\mathcal{N}_{sep} = O\left(m \log\left(\frac{mR}{\epsilon r}\right)\right)$ separation oracle calls.*

*Proof.* Let $\mathcal{N}_t$ be the number of times $t$ is updated; $\mathcal{N}_{in}$ the number of times $\mathcal{K}_{in}$ is updated; $\mathcal{N}_{out}$ the number of times $\mathcal{K}_{out}$ is updated; $\mathcal{N}_{\mathbf{x}}$ the number of times $\mathbf{x}$ is updated, and $\mathcal{N}_{total}$ the total number of iterations of the `while` loop before termination of Algorithm 1. Then, combining Lemma B.1, Lemma B.4, Lemma B.6, and Lemma B.7 gives

$$\Phi_{end} \leq \Phi_{init} + \mathcal{N}_{out} \cdot \log(1 - 1/e) + \mathcal{N}_t \cdot (\eta + \eta^2) + \mathcal{N}_{in} \cdot \log(1 - 1/e + \eta) + \mathcal{N}_{\mathbf{x}} \cdot \left(-\frac{\eta^2}{4}\right). \quad \text{(B.22)}$$

The initialization step of Algorithm 1 chooses $\eta = 1/100$, $t_{end} = \frac{8m}{\epsilon \|\mathbf{c}\|_2 R}$, and $t_{init} = \frac{m \log(m)}{\sqrt{n}\|\mathbf{c}\|_2 R}$, and we always update $t$ by a multiplicative factor of $1 + \frac{\eta}{4m}$ (see Line 9); therefore, we have

$$\mathcal{N}_t = O(m \log(mR/(\epsilon r))).$$

From Algorithm 1, the only times the separation oracle is invoked is when updating $\mathcal{K}_{in}$ or $\mathcal{K}_{out}$ in Line 15 and Line 17, respectively. Therefore, the total separation oracle complexity is $\mathcal{N}_{sep} = \mathcal{N}_{in} + \mathcal{N}_{out}$. Therefore, we have

$$\mathcal{N}_{sep} = \mathcal{N}_{in} + \mathcal{N}_{out} \leq O(1) \cdot [\Phi_{init} - \Phi_{end} + \mathcal{N}_t] = O(m \log(mR/(\epsilon r)))$$

This gives the claimed separation oracle complexity.

We now prove the guarantee on approximation. Let $\mathbf{x}_{output}$ be the output of Algorithm 1 and $\mathbf{x}$ be the point which entered Line 5 right before termination. Note that the termination of Algorithm 1 implies, by Line 5, that

$$\mathbf{c}^\top \mathbf{x}_{output} \leq \mathbf{c}^\top \mathbf{x} + \frac{\nu}{t_{end}} \leq \mathbf{c}^\top \mathbf{x}_{out}^\star + \frac{4(n+m)}{t_{end}} \leq \min_{\mathbf{x} \in \mathcal{K}, A\mathbf{x} = \mathbf{b}} \mathbf{c}^\top \mathbf{x} + \epsilon \cdot \|\mathbf{c}\|_2 \cdot R$$

where the first step is by the second inequality in Lemma C.8 (using the universal barrier) and the last step follows by our choice of $t_{end}$ and the definition of $\mathbf{x}_{out}^\star$ and $\mathcal{K}_{out} \supseteq \mathcal{K}$. $\square$

**Theorem B.10** (Main theorem of Problem 3.1). *Given the convex program*

$$\begin{aligned} \text{minimize} \quad & \langle \mathbf{c}, \mathbf{x} \rangle, \\ \text{subject to} \quad & \mathbf{x}_i \in \mathcal{K}_i \subseteq \mathbb{R}^{d_i+1} \forall i \in [n], \\ & A\mathbf{x} = \mathbf{b}. \end{aligned}$$

*Denote $\mathcal{K} = \mathcal{K}_1 \times \mathcal{K}_2 \times \ldots \times \mathcal{K}_n$. Assuming we have*

- *outer radius $R$: For any $\mathbf{x}_i \in \mathcal{K}_i$, we have $\|\mathbf{x}_i\|_2 \leq R$, and*

- *inner radius $r$: There exists a $\mathbf{z} \in \mathbb{R}^d$ such that $A\mathbf{z} = \mathbf{b}$ and $\mathcal{B}(\mathbf{z}, r) \subset \mathcal{K}$,*

*then, for any $0 < \epsilon < \frac{1}{2}$, we can find a point $\mathbf{x} \in \mathcal{K}$ satisfying $A\mathbf{x} = \mathbf{b}$ and*

$$\langle \mathbf{c}, \mathbf{x} \rangle \leq \min_{\substack{\mathbf{x}_i \in \mathcal{K}_i \subseteq \mathbb{R}^{d_i+1} \forall i \in [n], \\ A\mathbf{x} = \mathbf{b}}} \langle \mathbf{c}, \mathbf{x} \rangle + \epsilon \cdot \|\mathbf{c}\|_2 \cdot R,$$

*in $O(\operatorname{poly}(m \log(mR/\epsilon r)))$ time and using*

$$O(m \log(mR/(\epsilon r))$$

*gradient oracle calls, where $m = \sum_{i=1}^{n} d_i$.*

*Proof.* We apply Theorem C.1 for each $\mathcal{K}_i$ separately to find a solution $\mathbf{z}_i$. Then $\mathbf{z} = (\mathbf{z}_1, \ldots, \mathbf{z}_n) \in \mathbb{R}^{m+n}$ satisfies $\mathcal{B}_{m+n}(\mathbf{z}, \bar{r}) \subset \mathcal{K}$ with $\bar{r} = \frac{r}{6m^{3.5}}$. Then, we modified convex problem as in Definition C.5 with $s = 2^{16}\frac{m^{2.5}R}{r\epsilon}$ and obtaining the following:

$$
\begin{aligned}
\text{minimize} \quad & \langle \bar{\mathbf{c}}, \bar{\mathbf{x}} \rangle \\
\text{subject to} \quad & \bar{\mathbf{A}}\bar{\mathbf{x}} = \bar{\mathbf{b}}, \\
& \bar{\mathbf{x}} \in \bar{\mathcal{K}} := \mathcal{K} \times \mathbb{R}_{\geq 0}^{m+n} \times \mathbb{R}_{\geq 0}^{m+n}
\end{aligned}
\tag{B.23}
$$

with

$$
\bar{\mathbf{A}} = [\mathbf{A} \mid \mathbf{A} \mid -\mathbf{A}], \bar{\mathbf{b}} = \mathbf{b}, \bar{\mathbf{c}} = (\mathbf{c}, \frac{\|\mathbf{c}\|_2 s}{\sqrt{m+n}} \cdot \mathbf{1}, \frac{\|\mathbf{c}\|_2 s}{\sqrt{m+n}} \cdot \mathbf{1})^\top
$$

We solve the linear system $\mathbf{A}\mathbf{y} = \mathbf{b} - \mathbf{A}\mathbf{z}$ for $\mathbf{y}$. Then, we construct the initial $\bar{\mathbf{x}}$ by set $\bar{\mathbf{x}}^{(1)} = \mathbf{z}$,

$$
\bar{\mathbf{x}}_i^{(2)} = \begin{cases} \mathbf{y}_i & \text{if } \mathbf{y}_i \geq 0, \\ 0 & \text{otherwise.} \end{cases} \quad \text{and} \quad \bar{\mathbf{x}}_i^{(3)} = \begin{cases} -\mathbf{y}_i & \text{if } \mathbf{y}_i < 0, \\ 0 & \text{otherwise.} \end{cases}
$$

Then, we run Algorithm 1 on the Problem B.23, with initial $\bar{\mathbf{x}}$ set above, $\bar{m} = 3(m+n), \bar{n} = n+2, \bar{\epsilon} = \frac{\epsilon}{6\sqrt{n}s}, \overline{\mathcal{K}}_{\text{in}} = \{\mathbf{x}^{(1)} \in B(\mathbf{z}, \bar{r}), (\mathbf{x}^{(2)}, \mathbf{x}^{(3)}) \in \mathbb{R}_{\geq 0}^{2n}\}$ and $\widehat{\mathcal{K}}_{\text{out}} = \mathcal{B}_{\bar{m}}(\mathbf{0}, \sqrt{n}R)$.

By our choice of $t_{\text{end}}$, we have

$$
\bar{t}_{\text{end}} = \frac{8\bar{m}}{\bar{\epsilon}\|\bar{\mathbf{c}}\|_2 \bar{R}} \leq \frac{48m}{\epsilon\|\mathbf{c}\|_2 R}.
$$

First, we check the condition that $s \geq 48\bar{\nu}\bar{t}_{\text{end}}\sqrt{m+n}\frac{R^2}{r}\|\mathbf{c}\|_2$, we note that

$$
48\bar{\nu}\bar{t}_{\text{end}}\sqrt{m+n}\frac{R^2}{r}\|\mathbf{c}\|_2 \leq 27648\frac{m^{2.5}R}{\epsilon r} \leq 2^{16}\frac{m^{2.5}R}{r\epsilon} = s.
$$

Let $\bar{\mathbf{x}}_{output} = (\mathbf{x}_{output}^{(1)}, \mathbf{x}_{output}^{(2)}, \mathbf{x}_{output}^{(3)})$ be the output of Algorithm 1. Then, let $\mathbf{x}_{output} = \mathbf{x}_{output}^{(1)} + \mathbf{x}_{output}^{(2)} - \mathbf{x}_{output}^{(3)}$ as defined in Theorem C.6. By Lemma B.9, we have

$$
\min_{\mathbf{x} \in \mathcal{P}_{\text{in}}} \bar{\mathbf{c}}^\top \bar{\mathbf{x}} \leq \min_{\mathbf{x} \in \mathcal{P}} \mathbf{c}^\top \bar{\mathbf{x}} + \gamma
$$

where $\gamma = \bar{\epsilon} \cdot \|\bar{\mathbf{c}}\|_2 \cdot \bar{R}$.

Applying (3) of Theorem C.6, we have

$$
\mathbf{c}^\top \mathbf{x}_{output} \leq \frac{\bar{\nu}+1}{\bar{t}_{\text{end}}} + \gamma + \min_{x \in \mathcal{K}, A\mathbf{x}=\mathbf{b}} \mathbf{c}^\top \mathbf{x} \leq \min_{x \in \mathcal{K}, A\mathbf{x}=\mathbf{b}} \mathbf{c}^\top \mathbf{x} + \epsilon \cdot \|\mathbf{c}\|_2 \cdot R.
$$

The last inequality follows by our choice of $\bar{\epsilon}$ and $\bar{t}_{\text{end}}$, we have $\gamma \leq \frac{\epsilon}{2}\|\mathbf{c}\|_2 R$ and $\frac{\bar{\nu}+1}{\bar{t}_{\text{end}}} \leq \frac{\epsilon}{2}\|\mathbf{c}\|_2 R$. Plug this $\bar{\epsilon}$ in Lemma B.9, it gives the claimed oracle complexity.

$\square$

**Theorem 1.2** (Main Result). *Given Problem 1 and $\theta^{(0)}$ such that $\|\theta^\star - \theta^{(0)}\|_2 \leq R$. Assuming all the $f_i$'s are $L$-Lipschitz, then there is an algorithm that in time $\text{poly}(m\log(1/\epsilon))$, using $O(m\log(m/\epsilon))$ gradient oracle calls, outputs a vector $\theta \in \mathbb{R}^d$ such that*

$$
\sum_{i=1}^n f_i(\theta) \leq \sum_{i=1}^n f_i(\theta^\star) + \epsilon \cdot LR.
$$

*Proof.* First, we reformulate (1.1) using a change of variables and the epigraph trick. Suppose each $f_i$ depends on $d_i$ coordinates of $\theta$ given by $\{i_1, \ldots, i_{d_i}\} \subseteq [d]$. Then, symbolically define $\mathbf{x}_i = [x_{i_1}^{(i)}; x_{i_2}^{(i)}; \ldots; x_{i_{d_i}}^{(i)}] \in \mathbb{R}^{d_i}$ for each $i \in [n]$. Since each $f_i$ is convex and supported on $d_i$ variables, its epigraph is convex and $d_i + 1$ dimensional. So we may define the convex set

$$
\mathcal{K}_i^{\text{unbounded}} = \left\{ (\mathbf{x}_i, z_i) \in \mathbb{R}^{d_i+1} : f_i(\mathbf{x}_i) \leq Lz_i \right\}.
$$

Finally, we add linear constraints of the form $x_k^{(i)} = x_k^{(j)}$ for all $i, j, k$ where $f_i$ and $f_j$ both depend on $\theta_k$. We denote these by the matrix constraint $\mathbf{Ax} = \mathbf{b}$. Then, Problem 1 is equivalent to

$$
\begin{array}{ll}
\text{minimize} & \sum_{i=1}^{n} L z_i \\
\text{subject to} & \mathbf{Ax} = \mathbf{b} \\
& (\mathbf{x}_i, z_i) \in \mathcal{K}_i^{\text{unbounded}} \text{ for each } i \in [n].
\end{array}
\tag{B.24}
$$

Since we are given $\theta^{(0)}$ satisfying $\|\theta^{(0)} - \theta^*\|_2 \leq R$, we define $\mathbf{x}_i^{(0)} = [\theta_{i_1}^{(0)}; \ldots, \theta_{i_{d_i}}^{(0)}]$ and $z_i^{(0)} = f_i(\theta^{(0)})/L$. Then, we can restrict the search space $\mathcal{K}_i^{\text{unbounded}}$ to

$$
\mathcal{K}_i = \mathcal{K}_i^{\text{unbounded}} \cap \{(\mathbf{x}_i, z_i) \in \mathbb{R}^{d_i+1} : \|\mathbf{x}_i - \mathbf{x}_i^{(0)}\|_2 \leq R \text{ and } z_i^{(0)} - 2R \leq z_i \leq z_i^{(0)} + 2R\}.
$$

It's easy to check that $\mathcal{K}_i$ is contained in a ball of radius $5R$ centered at $(\mathbf{x}_i^{(0)}, z_i^{(0)})$, and contains a ball of radius $R$ centered at $(\mathbf{x}_i^{(0)}, z_i^{(0)})$. The subgradient oracle for $f_i$ translates to a separation oracle for $\mathcal{K}_i$. Then, we apply Theorem B.10 to (B.24) with $\mathcal{K}_i^{\text{unbounded}}$ replaced by $\mathcal{K}_i$ to get the error guarantee and oracle complexity directly. $\qquad\square$

Finally, we have the matching lower bound.

**Theorem 1.3.** *There exist functions $f_1, \ldots, f_n : \mathbb{R}^d \mapsto \mathbb{R}$ for which a total of $\Omega(m \log(1/\epsilon))$ gradient queries are required to solve Problem 1.*

*Proof.* [40] shows that for any $d_i$, there exists $f_i : \mathbb{R}^{d_i} \mapsto \mathbb{R}$ for which $\Omega(d_i \log(1/\epsilon))$ total gradient queries are required. We define $f_1, \ldots, f_n$ to be such functions on disjoint coordinates of $\theta$. It follows that $\Omega(\sum_{i=1}^{n} d_i \log(1/\epsilon)) = \Omega(m \log(1/\epsilon))$ gradient queries are required in total. $\qquad\square$

## C  Initialization

### C.1  Constructing an initial $\mathcal{K}_{\text{in},i}$

In this section, we discuss how to construct an initial set $\mathcal{K}_{\text{in},i}$ to serve as an input to Algorithm 1. In particular, we will prove the following theorem.

**Theorem C.1.** *Suppose we are given separation oracle access to a convex set $\mathcal{K}$ that satisfies $\mathcal{B}(\mathbf{z}, r) \subseteq \mathcal{K} \subseteq \mathcal{B}(\mathbf{0}, R)$ for some $\mathbf{z} \in \mathbb{R}^d$. Then, Algorithm 2, in $O(d \log(R/r))$ separation oracle calls to $\mathcal{K}$, outputs a point $\mathbf{x}$ such that $\mathcal{B}\left(\mathbf{x}, \frac{r}{6d^{3.5}}\right) \subseteq \mathcal{K}$.*

---

**Algorithm 2** Inner Ball Finding

1: $\mathcal{K}_{\text{out}} \leftarrow B(0, R)$
2: **while true do**
3: $\quad$ Let $\mathbf{v}$ be the center of gravity of $\mathcal{K}_{\text{out}}$
4: $\quad$ Sample $\mathbf{u}$ from $B(\mathbf{v}, r/(6d))$ uniformly
5: $\quad$ **if** $\mathbf{u} \in \mathcal{K}$ **then**
6: $\quad\quad$ Let $S = \{\mathbf{v} \pm \frac{r}{6d^3}\mathbf{e}_i : i \in [d]\}$
7: $\quad\quad$ **if** $S \subset \mathcal{K}$ **then**
8: $\quad\quad\quad$ **return** the inscribed ball of $\text{conv}(S)$
9: $\quad\quad$ **end if**
10: $\quad$ **end if**
11: $\quad$ Let $\mathcal{K}_{\text{out}} \leftarrow \mathcal{K}_{\text{out}} \cap \mathcal{H}$ where $\mathcal{H} = \mathcal{O}(\mathbf{u})$
12: **end while**

---

Before we prove the preceding theorem, we need the following facts about the self-concordant barrier and convex sets.

**Theorem C.2** ([40, Theorem 4.2.6]). *Let $\psi : int(\mathcal{K}) \to \mathbb{R}$ be a $\nu$-self-concordant barrier with the minimizer $\mathbf{x}_\psi^\star$. Then, for any $\mathbf{x} \in int(\mathcal{K})$ we have:*

$$\|\mathbf{x}_\psi^\star - \mathbf{x}\|_{\mathbf{x}_\psi^\star} \le \nu + 2\sqrt{\nu}.$$

*On the other hand, for any $\mathbf{x} \in \mathbb{R}^d$ such that $\|\mathbf{x} - \mathbf{x}_\psi^\star\|_{\mathbf{x}_\psi^\star} \le 1$, we have $x \in int(\mathcal{K})$.*

**Theorem C.3** ([26, Theorem 4.1]). *Let $\mathcal{K} \subseteq \mathbb{R}^d$ be a convex set with center of gravity $\mu$ and covariance matrix $\Sigma$. Then,*

$$\{\mathbf{x} : \|\mathbf{x} - \mu\|_{\Sigma^{-1}} \le \sqrt{(d+2)/d}\} \subseteq \mathcal{K} \subseteq \{\mathbf{x} : \|\mathbf{x} - \mu\|_{\Sigma^{-1}} \le \sqrt{d(d+2)}\}.$$

**Theorem C.4** ([9, Section 1.4.2]). *Let $\mathcal{K}$ be a convex set with $\mathcal{K} \subset B(\mathbf{u}, R)$ for some $R$. Let $\mathcal{K}_{-\delta} = \{\mathbf{x} : B(\mathbf{x}, \delta) \subset \mathcal{K}\}$. Then, we have*

$$\mathrm{vol}\mathcal{K}_{-\delta} \ge \mathrm{vol}\mathcal{K} - (1 - (1 - \frac{\delta}{R})^d) \cdot \mathrm{vol}B(\mathbf{u}, R)$$

*Proof of Theorem C.1.* We note that by the description of the Algorithm 2, the returned ball is the inscribed ball of $\mathrm{conv}(S)$ and we have $\mathbf{v} \in \mathcal{K}$ for each $\mathbf{v} \in S$. Then, we must have $\mathrm{conv}(S) \subseteq \mathcal{K}$. We note that $\mathrm{conv}(S)$ is a $\ell_1$ ball with $\ell_1$ radius $\frac{r}{6d^3}$, then the inscribed ball has $\ell_2$ radius $\frac{r}{6d^{3.5}}$.

First, we prove the sample complexity of the algorithm above. We use $\mathcal{K}_t$ to denote the $\mathcal{K}_{\mathrm{out}}$ at the $t$-th iteration. We first observe that throughout the algorithm, $\mathcal{K}_t$ is obtained by intersection of halfspaces and $B(0, R)$. This implies

$$B(\mathbf{z}, r) \subseteq \mathcal{K} \subseteq \mathcal{K}_t \qquad \forall t.$$

Since $\mathcal{K}_t$ contains a ball of radius $r$, let $A_t$ be the covariance matrix of $\mathcal{K}_t$. By Theorem C.3, we have

$$A_t \succeq \frac{r^2}{d(d+2)} I.$$

Let $\mathcal{H}_t$ be the halfspace returned by the oracle at iteration $t$. We note that $\mathbf{u}$ is sampled uniform from $B(\mathbf{v}, r/(6d))$, so we have

$$\|\mathbf{v} - \mathbf{u}\|_{A^{-1}} \le \frac{\sqrt{d(d+2)}}{r} \cdot \frac{r}{6d} \le \frac{1}{3}.$$

Apply the inequality above to Corollary A.10, we have

$$\mathrm{vol}(\mathcal{K}_t) \le (1 - 1/e + 1/3)^t \mathrm{vol}(\mathcal{K}_0) \le (1 - 1/30)^t \mathrm{vol}(B(0, R)).$$

Then, since $B(\mathbf{z}, r) \subseteq \mathcal{K}_t$ for all the $t$, this implies the algorithm at most takes $O(d \log(R/r))$ many iterations.

Now, we consider the number of oracle calls within each iterations. There are three possible cases to consider:

1. $\mathbf{u} \in \mathcal{K}_{-\delta}$ with $\delta = \frac{r}{6d^3}$ (see the definition of $\mathcal{K}_{-\delta}$ in Theorem C.4). In this case, we have $S \subset \mathcal{K}$ and this is the last iteration. We can pay this $O(d)$ oracle calls for the last iteration.

2. $\mathbf{u} \in \mathcal{K} \backslash \mathcal{K}_{-\delta}$.

   Since $\mathbf{u}$ is uniformly sampled from $B(\mathbf{v}, r/(6d))$, Theorem C.4 shows that $\mathbf{u} \in \mathcal{K} \backslash \mathcal{K}_{-\delta}$ with probability at most

   $$1 - (1 - \frac{\delta}{r/(6d)})^d \le \frac{1}{d}.$$

   Hence, this case only happens with probability only at most $1/d$. Since the cost of checking $S \subset \mathcal{K}$ takes $O(d)$ oracle calls. The expected calls for this case is only $O(1)$.

3. $\mathbf{u} \notin K$. The cost is just 1 call.

Combining all the cases, the expected calls is $O(1)$ per iteration.

$\square$

## C.2 Initial point reduction

In this section, we will show how to obtain an initial feasible point for the algorithm.

**Definition C.5.** Given a convex program $\min_{\mathbf{A}\mathbf{x}=\mathbf{b},\mathbf{x}\in\mathcal{K}\subseteq\mathbb{R}^d} \mathbf{c}^\top\mathbf{x}$ and some $s > 0$, we define $\mathbf{c}_1 = \mathbf{c}, \mathbf{c}_2 = \mathbf{c}_3 = \frac{s\|\mathbf{c}\|_2}{\sqrt{d}} \cdot \mathbf{1}$ and $\mathcal{P} = \{\mathbf{x}^{(1)} \in \mathcal{K}, (\mathbf{x}^{(2)}, \mathbf{x}^{(3)}) \in \mathbb{R}^{2d}_{\geq 0} : \mathbf{A}(\mathbf{x}^{(1)} + \mathbf{x}^{(2)} - \mathbf{x}^{(3)}) = \mathbf{b}\}$. We then define the *modified convex program* by

$$\min_{(\mathbf{x}^{(1)},\mathbf{x}^{(2)},\mathbf{x}^{(3)})\in\mathcal{P}} \mathbf{c}_1^\top\mathbf{x}^{(1)} + \mathbf{c}_2^\top\mathbf{x}^{(2)} + \mathbf{c}_3^\top\mathbf{x}^{(3)}.$$

We denote $(\mathbf{c}_1, \mathbf{c}_2, \mathbf{c}_3)$ by $\bar{\mathbf{c}}$.

**Theorem C.6.** *Given a convex program $\min_{\mathbf{A}\mathbf{x}=\mathbf{b},\mathbf{x}\in\mathcal{K}\subseteq\mathbb{R}^d} \mathbf{c}^\top\mathbf{x}$ with outer radius $R$ and some convex set $\mathcal{K}_{in}$ with $\mathcal{K}_{in} \subseteq \mathcal{K}$ and inner radius $r$. For any modified convex program as in Definition C.5 with $s \geq 48\nu t\sqrt{d} \cdot \frac{R}{r} \cdot \|c\|_2 R$. For an arbitrary $t \in \mathbb{R}_{\geq 0}$, we define the function*

$$f_t(\mathbf{x}^{(1)}, \mathbf{x}^{(2)}, \mathbf{x}^{(3)}) = t(\mathbf{c}_1^\top\mathbf{x}^{(1)} + \mathbf{c}_2^\top\mathbf{x}^{(2)} + \mathbf{c}_3^\top\mathbf{x}^{(3)}) + \psi_{\mathcal{P}_{in}}(\mathbf{x}^{(1)}, \mathbf{x}^{(2)}, \mathbf{x}^{(3)})$$

*where $\psi_{\mathcal{P}_{in}}$ is some $\nu$ self-concordant barrier for the set*

$$\mathcal{P}_{in} = \{\mathbf{x}^{(1)} \in \mathcal{K}_{in}, (\mathbf{x}^{(2)}, \mathbf{x}^{(3)}) \in \mathbb{R}^{2d}_{\geq 0} : \mathbf{A}(\mathbf{x}^{(1)} + \mathbf{x}^{(2)} - \mathbf{x}^{(3)}) = \mathbf{b}\}.$$

*Given $\bar{\mathbf{x}}_t := (\mathbf{x}_t^{(1)}, \mathbf{x}_t^{(2)}, \mathbf{x}_t^{(3)}) = \arg\min_{(\mathbf{x}^{(1)},\mathbf{x}^{(2)},\mathbf{x}^{(3)})\in\mathcal{P}_{in}} f_t(\mathbf{x}^{(1)}, \mathbf{x}^{(2)}, \mathbf{x}^{(3)})$, we denote $\mathbf{x}_{in} = \mathbf{x}_t^{(1)} + \mathbf{x}_t^{(2)} - \mathbf{x}_t^{(3)}$. Suppose $\min_{\bar{\mathbf{x}}\in\mathcal{P}_{in}} \bar{\mathbf{c}}^\top\bar{\mathbf{x}} \leq \min_{\bar{\mathbf{x}}\in\mathcal{P}} \bar{\mathbf{c}}^\top\bar{\mathbf{x}} + \gamma$, we have the following*

1. $\mathbf{A}\mathbf{x}_{in} = \mathbf{b}$,

2. $\mathbf{x}_{in} \in \mathcal{K}_{in}$,

3. $\mathbf{c}^\top\mathbf{x}_{in} \leq \min_{\mathbf{x}\in\mathcal{K},\mathbf{A}\mathbf{x}=\mathbf{b}} \mathbf{c}^\top\mathbf{x} + \frac{\nu+1}{t} + \gamma$.

First, we show that $\mathbf{x}_t^{(1)}$ is not too close to the boundary. Before we proceed, we need the following lemmas.

**Lemma C.7** (Theorem 4.2.5 [40]). *Let $\psi$ be a $\nu$-self-concordant barrier. Then, for any $\mathbf{x} \in dom(\psi)$ and $\mathbf{y} \in dom(\psi)$ such that*

$$\langle \psi'(\mathbf{x}), \mathbf{y} - \mathbf{x} \rangle \geq 0,$$

*we have*

$$\|\mathbf{y} - \mathbf{x}\|_{\mathbf{x}} \leq \nu + 2\sqrt{\nu}.$$

**Lemma C.8** (Theorem 2 of [57]). *Given a convex set [5] $\Omega$ with a $\nu$-self-concordant barrier $\psi_\Omega$ and inner radius $r$. Let $\mathbf{x}_t = \arg\min_{\mathbf{x}} t \cdot \mathbf{c}^\top\mathbf{x} + \psi_\Omega(\mathbf{x})$. Then, for any $t > 0$,*

$$\min\left\{\frac{1}{2t}, \frac{r\|c\|_2}{4\nu + 4\sqrt{\nu}}\right\} \leq \mathbf{c}^\top\mathbf{x}_t - \mathbf{c}^\top\mathbf{x}_\infty \leq \frac{\nu}{t}.$$

Consider the optimization problem restricted in the subspace $\{(\mathbf{x}^{(1)}, \mathbf{x}^{(2)}, \mathbf{x}^{(3)}) : A(\mathbf{x}^{(1)} + \mathbf{x}^{(2)} - \mathbf{x}^{(3)}) = \mathbf{b}\}$, as a direct corollary of theorem above we have the following:

**Corollary C.9.** *Let $\bar{\mathbf{x}}_t$ be as the same as defined in Theorem C.6. For $t \geq \frac{4\nu}{r\|c\|_2}$, we have $dist(\mathbf{x}_t^{(1)}, \mathbf{x}_\infty^{(1)}) \geq \frac{1}{2t\|\mathbf{c}\|_2}$.*

Now, we are ready to show $dist(\mathbf{x}_t^{(1)}, \partial\mathcal{K}_{in})$ is not too small.

**Theorem C.10.** *Let $\bar{\mathbf{x}}_t$ be the same as defined in Theorem C.6. For $t \geq \frac{4\nu}{r\|c\|_2}$, we have $dist(\mathbf{x}_t^{(1)}, \partial\mathcal{K}_{in}) \geq \frac{r}{12\nu t\|\mathbf{c}\|_2 R}$.*

---

[5]The original theorem is stated only for polytopes, but their proof works for general convex sets.

*Proof.* We consider the domain restricted in the subspace $\{(\mathbf{x}^{(1)}, \mathbf{x}^{(2)}, \mathbf{x}^{(3)}) : \mathbf{A}(\mathbf{x}^{(1)} + \mathbf{x}^{(2)} - \mathbf{x}^{(3)}) = \mathbf{b}\}$. By the optimality of $\bar{\mathbf{x}}_t$ and Lemma C.7, we have

$$\mathcal{K}_{\mathcal{H}} \subseteq \{\mathbf{x} : \|\mathbf{x} - \mathbf{x}_t^{(1)}\|_{\mathbf{x}_t^{(1)}} \le \nu + 2\sqrt{\nu}\},$$

where $\mathcal{H} = \{\mathbf{x} : \mathbf{c}^\top(\mathbf{x}_t^{(1)} - \mathbf{x}) \ge 0\}$ and $\mathcal{K}_{\mathcal{H}} := \mathcal{H} \cap \mathcal{K}_{\text{in}}$.

Recall that $\mathcal{K}_{\text{in}}$ contains a ball of radius $r$, we denote it by $B$. We note that $\text{conv}(\mathbf{x}_\infty^{(1)}, B)$ is a union of a ball and a convex cone $\mathcal{C}$ with diameter at most $2R$. We observe that the set $\text{conv}(\mathbf{x}_\infty^{(1)}, B) \cap \mathcal{H}$ contains a ball of radius at least $\frac{r}{4t\|\mathbf{c}\|_2 R}$ since $\text{dist}(\mathbf{x}_\infty^{(1)}, \partial\mathcal{H}) \ge \frac{1}{2t\|\mathbf{c}\|_2}$.

We note that

$$\text{conv}(\mathbf{x}_\infty^{(1)}, B) \cap \mathcal{H} \subseteq \mathcal{K}_{\text{in}} \subseteq \{\mathbf{x} : \|\mathbf{x} - \mathbf{x}_t^{(1)}\|_{\mathbf{x}_t^{(1)}} \le \nu + 2\sqrt{\nu}\},$$

this implies $\{\mathbf{x} : \|\mathbf{x} - \mathbf{x}_t^{(1)}\|_{\mathbf{x}_t^{(1)}} \le \nu + 2\sqrt{\nu}\}$ contains a ball of radius at least $\frac{r}{4t\|\mathbf{c}\|_2 R}$, and then by Theorem C.2, we have $B(\mathbf{x}_t^{(1)}, \frac{r}{4(\nu + 2\sqrt{\nu})t\|\mathbf{c}\|_2 R}) \subseteq \mathcal{K}_{\text{in}}$. $\square$

**Lemma C.11.** *Let* $(\mathbf{x}_t^{(1)}, \mathbf{x}_t^{(2)}, \mathbf{x}_t^{(3)}) \in \mathbb{R}^{3d}$ *be the same as defined in Theorem C.6. If* $t > \frac{\nu}{\|\mathbf{c}\|_2 R}$, *then we have* $\|\mathbf{x}_t^{(2)} - \mathbf{x}_t^{(3)}\|_2 \le \frac{4\sqrt{d}}{s} R$.

*Proof.* Let $\mathbf{x}_{\text{in}}^\star = \arg\min_{\mathbf{x} \in \mathcal{K}_{\text{in}}, \mathbf{A}\mathbf{x}=\mathbf{b}} \mathbf{c}^\top \mathbf{x}$ and $\overline{\mathbf{x}}_{\text{in}}^\star = \arg\min_{\mathbf{x} \in \mathcal{P}_{\text{in}}} \overline{\mathbf{c}}^\top \overline{\mathbf{x}}$. Since $\mathbf{x}^\star \in \mathcal{B}(0, R)$, we have

$$\mathbf{c}^\top \mathbf{x}_{\text{in}}^\star \le \|\mathbf{c}\|_2 R.$$

Note that $(\mathbf{x}_{\text{in}}^\star, \mathbf{0}, \mathbf{0}) \in \mathcal{P}_{\text{in}}$, this means we have

$$\overline{\mathbf{c}}^\top \overline{\mathbf{x}}_{\text{in}}^\star \le \mathbf{c}^\top \mathbf{x}_{\text{in}}^\star \le \|\mathbf{c}\|_2 R.$$

Combining this with the second inequality in Lemma C.8, we get

$$\overline{\mathbf{c}}^\top \overline{\mathbf{x}}_t \le \overline{\mathbf{c}}^\top \overline{\mathbf{x}}_{\text{in}}^\star + \frac{\nu}{t} \le \|\mathbf{c}\|_2 R + \frac{\nu}{t} \le 2\|\mathbf{c}\|_2 R.$$

We further note that

$$\mathbf{c}_2^\top \mathbf{x}_t^{(2)} \le \overline{\mathbf{c}}^\top \overline{\mathbf{x}}_t \le 2\|\mathbf{c}\| R.$$

This shows

$$\max\{\|\mathbf{x}_t^{(2)}\|_2, \|\mathbf{x}_t^{(3)}\|_2\} \le \frac{2\sqrt{d}\|\mathbf{c}\|_2 R}{\|\mathbf{c}\|_2 s} \le \frac{2\sqrt{d}R}{s}.$$

Hence, we have

$$\|\mathbf{x}_t^{(2)} - \mathbf{x}_t^{(3)}\|_2 \le \frac{4\sqrt{d}}{s} R.$$

$\square$

Now, we are ready to prove Theorem C.6.

*Proof of Theorem C.6.* We note that $\mathbf{x}_{\text{in}}$ satisfies (1), directly follows by definition of $\mathcal{P}$. By assumption, we have $s \ge 48\nu t\sqrt{d} \cdot \frac{R}{r} \cdot \|c\|_2 R$; using this in Lemma C.11, we have

$$\|\mathbf{x}_t^{(2)} - \mathbf{x}_t^{(3)}\|_2 \le \frac{r}{12\nu t\|\mathbf{c}\|_2 R}.$$

This means $\mathbf{x}_{\text{in}} = \mathbf{x}_t^{(1)} + \mathbf{x}_t^{(2)} - \mathbf{x}_t^{(3)} \in \mathcal{K}_{\text{in}}$ since $\text{dist}(\mathbf{x}_t^{(1)}, \partial\mathcal{K}_{\text{in}}) \ge \frac{r}{12\nu t\|\mathbf{c}\|_2 R}$.

Now, we show $\mathbf{c}^\top \mathbf{x}_{\text{in}}$ is close to $\mathbf{c}^\top \mathbf{x}^\star$.

Let $\mathbf{x}^\star = \arg\min_{\mathbf{x} \in \mathcal{K}, \mathbf{A}\mathbf{x}=\mathbf{b}} \mathbf{c}^\top \mathbf{x}$ and $\overline{\mathbf{x}}^\star = \arg\min_{\mathbf{x} \in \mathcal{P}} \overline{\mathbf{c}}^\top \overline{\mathbf{x}}$. By Lemma C.8, we have

$$\overline{\mathbf{c}}^\top \overline{\mathbf{x}}_t - \frac{\nu}{t} \le \overline{\mathbf{c}}^\top \overline{\mathbf{x}}_{\text{in}}^\star \le \overline{\mathbf{c}}^\top \overline{\mathbf{x}}^\star + \gamma \le \mathbf{c}^\top \mathbf{x}^\star + \gamma.$$

This implies

$$\mathbf{c}^\top \mathbf{x}_t^{(1)} \leq \overline{\mathbf{c}}^\top \overline{\mathbf{x}}_t \leq \mathbf{c}^\top \mathbf{x}^\star + \frac{\nu}{t} + \gamma.$$

We have

$$\mathbf{c}^\top \mathbf{x}_{\text{in}} = \mathbf{c}^\top (\mathbf{x}_t^{(1)} + \mathbf{x}_t^{(2)} - \mathbf{x}_t^{(3)}) \leq \mathbf{c}^\top \mathbf{x}^\star + \frac{\nu}{t} + \frac{4}{s}\|\mathbf{c}\|_2 R \leq \mathbf{c}^\top \mathbf{x}^\star + \frac{\nu + 1}{t} + \gamma.$$

$\square$

## D  Decomposable submodular function minimization

### D.1  Preliminaries

Throughout, $V$ denotes the ground set of elements. A set function $f : 2^V \to \mathbb{R}$ is *submodular* if it satisfies the following *diminishing marginal differences* property:

**Definition D.1** (Submodularity). A function $f : 2^V \to \mathbb{R}$ is submodular if $f(T \cup \{i\}) - f(T) \leq f(S \cup \{i\}) - f(S)$, for any subsets $S \subseteq T \subseteq V$ and $i \in V \setminus T$.

We may assume without loss of generality that $f(\emptyset) = 0$ by replacing $f(S)$ by $f(S) - f(\emptyset)$. We assume that $f$ is accessed by an *evaluation oracle* and use EO to denote the time to compute $f(S)$ for a subset $S$. Our algorithm for decomposable SFM is based on the Lovász extension [19], a standard convex extension of a submodular function.

**Definition D.2** (Lovász extension [19]). The Lovász extension $\hat{f} : [0,1]^V \to \mathbb{R}$ of a submodular function $f$ is defined as

$$\hat{f}(x) = \mathbb{E}_{t \sim [0,1]}[f(\{i \in V : x_i \geq t\})],$$

where $t \sim [0,1]$ is drawn uniformly at random from $[0,1]$.

The Lovász extension $\hat{f}$ of a submodular function $f$ has many desirable properties. In particular, $\hat{f}$ is a convex relaxation of $f$ and it can be evaluated efficiently.

**Theorem D.3** (Properties of Lovász extension [19]). *Let $f : 2^V \to \mathbb{R}$ be a submodular function and $\hat{f}$ be its Lovász extension. Then,*

*(a)* $\hat{f}$ *is convex and* $\min_{x \in [0,1]^V} \hat{f}(x) = \min_{S \subseteq V} f(S)$*;*

*(b)* $f(S) = \hat{f}(I_S)$ *for any subset $S \subseteq V$, where $I_S$ is the indicator vector for $S$;*

*(c) Suppose $x \in [0,1]^V$ satisfies $x_1 \geq \cdots \geq x_{|V|}$, then $\hat{f}(x) = \sum_{i=1}^{|V|}(f([i]) - f([i-1]))x_i$.*

Property (c) in Theorem D.3 allows us to implement a sub-gradient oracle for $\hat{f}$ by evaluating $f$.

**Theorem D.4** (Sub-gradient oracle implementation for Lovász extension, Theorem 61 of [33]). *Let $f : 2^V \to \mathbb{R}$ be a submodular function and $\hat{f}$ be its Lovász extension. Then a sub-gradient for $\hat{f}$ can be implemented in time $O(|V| \cdot \text{EO} + |V|^2)$.*

### D.2  Decomposable submodular function minimization proofs

In this subsection, we prove the following more general version of Theorem 1.4.

**Theorem D.5** (Decomposable SFM). *Let $F : V \to [-1,1]$ be given by $F(S) = \sum_{i=1}^n F_i(S \cap V_i)$, where each $F_i : 2^{V_i} \to \mathbb{R}$ is a submodular function on $V_i \subseteq V$ with $|V_i| = d_i$. Let $m = \sum_{i=1}^n d_i$ and $d_{\max} := \max_{i \in [n]} d_i$. Then we can find an $\epsilon$-approximate minimizer of $f$ using at most $O(d_{\max} m \log(m/\epsilon))$ evaluation oracle calls.*

*Proof.* Let $\hat{f}_i$ be the Lovász extension of each $f_i$, then $\hat{f} = \sum_{i=1}^n \hat{f}_i$ is the Lovász extension of $f$. Note that $\hat{f}$ is 2-Lipschitz since the range of $f$ is $[-1,1]$. Also, the diameter of the range $[0,1]^{V_i}$ for each Lovász extension $\hat{f}_i$ is at most $\sqrt{|V_i|} \leq \sqrt{d_{\max}}$. Thus using Theorem 1.2, we can find a vector $x \in [0,1]^V$ such that $\hat{f}(x) \leq \min_{x^* \in [0,1]^V} \hat{f}(x^*) + \epsilon$ in $\text{poly}(m \log(1/\epsilon))$ time

and $O(m \log(m\sqrt{d_{\max}}/\epsilon)) = O(m \log(m/\epsilon))$ subgradients of the $\hat{f}_i$'s. By Theorem D.4, each sub-gradient of $\hat{f}_i$ can be computed by making at most $d_i \leq d_{\max}$ queries to the evaluation oracle for $f_i$. Thus the total number of evaluation oracle calls we make in finding an $\epsilon$-additive approximate minimizer $x \in [0,1]^V$ of $\hat{f}$ is at most $O(d_{\max} m \log(m/\epsilon))$.

Next we turn the $\epsilon$-additive approximate minimizer $x$ of $\hat{f}$ into an $\epsilon$-additive approximate minimizer $S \subseteq V$ for $f$. Without loss of generality, assume that $x_1 \geq \cdots \geq x_{|V|}$. Then by property (c) in Theorem D.3, we have

$$\hat{f}(x) = \sum_{i=1}^{|V|} (f([i]) - f([i-1]))x_i = f(V) \cdot x_{|V|} + \sum_{i=1}^{|V|-1} f([i]) \cdot (x_i - x_{i+1}).$$

Since $x_i - x_{i+1} \geq 0$, the above implies that $\min_{i \in \{1,\ldots,|V|\}} f([i]) \leq \hat{f}(x)$. Thus we can find a subset $S \subseteq V$ among $f([i])$ for all $i \in \{1, \cdots, |V|\}$ such that $f(S) \leq \hat{f}(x)$. Then by property (a) in Theorem D.3, the set $S$ is an $\epsilon$-additive approximate minimizer of $f$. This proves the theorem. $\quad\square$