# OpenReview forum: "Decomposable Non-Smooth Convex Optimization with Nearly-Linear Gradient Oracle Complexity"
_NeurIPS.cc/2022/Conference — NeurIPS 2022 Accept_

### Official Review · Reviewer_TejV · 2022-07-04

**Rating:** 7
**Confidence:** 4
**Soundness:** 4 excellent
**Presentation:** 2 fair
**Contribution:** 4 excellent

**Summary:**

In this work, the authors proposed a weakly-polynomial algorithm to minimize the finite-sum objective function, which is possibly non-smooth. The complexity of the proposed algorithm improves that of the cutting-plane methods, under the assumption that each component f_i only depends on a small subset of the components of x. The algorithm is a combination of the interior-point method and the cutting-plane method.

**Questions:**

Major:
(1) The presentation of the paper should be improved. For example, the discussion in lns.140-149 is based on definitions/results in Section 3. The readers have to complete reading the whole paper to understand the intuition that the authors wanted to convey here. This will make this discussion less useful in "introducing" the readers to the contributions of this paper. It might be helpful to explain what are inequalities (3.3) and (3.4).

(2) In addition, it is a little confusing why the complexity of the proposed algorithm has dependence on m rather than nd. It seems to me that the dependence on m suddenly appeared on ln.325. Hence, it would be better to provide an intuition on the dependence on m.


Minor:
(1) It would be better to mention that in the SFM problem, one can utilize the Lovasz extension to compute a sub-gradient in O(d) steps. Otherwise, readers that didn't read the appendix will be confused on the relation between the evaluation oracle and the sub-gradient oracle.

(2) Ln.94: it would be better to clarify the definition of the "polynomial factor". Namely, is this factor a polynomial of d_i and epsilon, or something else?

(3) Ln.102: since the AGD algorithm updates one coordinate in each iteration, I wonder if the complexity is proportional to d? Also, I do not understand why AGD can only update one coordinate in each iteration, and it seems that the complexity Omega(n^2) does not match the complexity in Table 1? It would be better to provide more details to this discussion.

(4) Ln.106: it would be better to use "sub-gradient" instead of "gradient" in the discussion of CPM, since it does not require the smoothness condition.

(5) Ln.127: K_i is not defined.

(6) Ln.138: please unify the usage of "sub-gradient" and "subgradient". Also, "a separation oracle K_i" -> "a separation oracle of K_i".

(7) What are c, A and b in problems (1.4) and (3.1)?

(8) Lns.140-149: it seems that the case when inequality (3.4) is violated is not discussed.

(9) Ln.175: y should be boldface letter.

(10) Ln.200: I wonder if the convex set K_i is defined by the epigraph of f_i? If so, please clarify the meaning of "does not have a closed-form expression". If not, please provide a more explicit definition of K_i.

(11) Ln.224: does the Newton step refer to Ln.23 of Algorithm 1?

(12) For the discussion around Prop.2, I wonder what is the difference between the claim here, and the classic "volume reduction" results in CPM algorithms?

**Limitations:**

See my comments in the "questions" section.

**Strengths And Weaknesses:**

This work provides a novel algorithm for the non-smooth finite-sum optimization problem. In addition, the complexity is improved in certain cases. Although I haven't read the appendix, the intuitions on the technical proofs seem reasonable to me. Therefore, I think this work will be interesting to audiences in optimization and machine learning.

On the other side, given the technical complexities of this work and the page limit of the conference, it is a little hard to present the results clearly. With that said, I still suggest the authors try to revise the presentation and organization of this work. In addition, it would be better if numerical results can be provided.

---

> ### Author Response · Authors · 2022-08-01
> **Response to Reviewer TejV**
>
> We thank the reviewer for their careful review of our paper and for providing very insightful comments. We will incorporate the reviewer’s suggestions for improving the presentation in our next version. Below are our responses to other specific comments/questions.
>
> ### Dependence on $m$
> By reformulating the original problem (1.1) to problem (1.4), we obtain a problem with dimension $(m+n) \leq 2m$. When the functions $f_i$ depend on disjoint coordinates, there is no linear constraint (i.e. no $Ax = b$) and directly applying cutting plane methods would require $\widetilde{O}(m)$ gradient calls (this comes from the fact that each sub-gradient oracle call decreases the volume of the search space by a constant, or that cutting plane methods make roughly linear number of sub-gradient queries). This shows that intuitively we should hope for the oracle complexity to be nearly linear in the ambient dimension $m$.
>
> It turns out that this intuition is correct even when $f_i$ share coordinates (i.e. there are linear constraints in (1.4)), and showing this is one main novelty of our techniques.
>
> ### Response to minor comments
>
> 1. Indeed. We will mention this in the main text in our next version.
>
> 2. The polynomial factor in L94 is in $m$ and $\log (1/\epsilon)$. We will clarify this in our next version.
>
> 3. As shown in Ln49-54, the $\kappa$ can be as large as $n^2$, AGD takes $O(n\sqrt\kappa)$ and hence $\Omega(n^2)$ is consistent with the complexity in the table.
>
> 4. We totally agree and will modify it in our next version.
>
> 5. Currently $K_i$ is defined in the proof of Thm B.11 (Ln759-770). For better presentation, we will include its definition before ln 127 in our next version.
>
> 6. We will unify and fix these in our next version.
>
> 7. The definition of A,b and c are currently in the proof of Thm B.11 (Ln759-770). We plan to move it to around (1.4) in our next version.
>
> 8. If inequality 3.4 is violated, then it means the current point $x$ is already approximately optimal and we would be done. We will make this explicit in the next version of the paper.
>
> 9. We will fix this.
>
> 10. Yes the K_i is the epigraph. We meant to say that the subgradient oracle for f_i translates into a separation oracle for K_i, and this is the only way to access K_i. We will make this more explicit in our next version.
>
> 11. Yes
>
> 12. The argument about CPM is pretty much the same. We want to point out that the novelty of this paper is that we crucially use the fact that we are able to make progress in both cases. If the point is outside our convex body, our argument follows the standard volume argument of CPM, while if the point is inside the convex body, we are able to grow our inner convex approximation.
>
> We again thank the reviewer for their time and effort in reading the paper and for providing very insightful comments.

---

> > ### Comment · Reviewer_TejV · 2022-08-07
> > **Post-rebuttal**
> >
> > I would like to thank the authors for responding to my questions. I have read all reviews and responses. I echo reviewer 5YUr's concern on the implementation issue of the proposed algorithms. The authors should consider adding a discussion on the implementation techniques of these algorithms. With that said, given the correctness of the proofs (which I couldn't check due to the time limit), I feel that the theoretical results in this work should be novel and interesting to the audience of NeurIPS. In summary, I would like to keep my score but would not fight for the acceptance of this work.

---

> > > ### Author Response · Authors · 2022-08-07
> > > **Response to Post-rebuttal by Reviewer TejV**
> > >
> > > We thank the reviewer for their additional comment and echoing the concern about the implementation issue of the proposed algorithm. The following is our response to the implementation issue (this part is the same as our response to reviewer 5YUr's additional comments).
> > >
> > > ### Implementation Issue
> > > Note that the gradient and hessian of the entropy barrier are the centroid and covariance of some exponential distribution over a polytope. In our setting, we only need a constant approximation of the centroid and covariance. In general, sampling from log-concave distribution takes n^4.2 time, but in our case, sampling from an exponential distribution over explicit polytope is much easier --- It takes around quadratic time in practice if the polytope is sparse, which holds for our case because of our assumption that each f_i only depends on a few variables.
> > >
> > > These results (and runtime complexity) of sampling from an exponential distribution over explicit polytope are not purely theoretical. For example, one problem in Bioinformatics about simulating metabolic networks (which contains tens of thousands of variables to hundreds of thousands of variables) involves computing centroid and covariance of exponential distribution over polytopes. As a result, they have been inventing new software for this task. One popular package for this problem is https://opencobra.github.io/cobratoolbox/stable/. It can sample from a polytope with tens of thousands of variables currently in seconds. In this package, there are ACHR, CHRR, RHMC, representing three iterations of improvement on the sampler in last 20 years. For example, for one human model (RECON3) with 12k variables, ACHR probably takes 10^8 cpu-sec to obtain one sample from that polytope while CHRR takes 10^4 cpu-sec and RHMC takes 30 sec.
> > >
> > > Therefore, we think it is evident that there could be an implementation of our algorithm which can perform reasonably well in practice.

---

### Official Review · Reviewer_5YUr · 2022-07-11

**Rating:** 4
**Confidence:** 3
**Soundness:** 3 good
**Presentation:** 3 good
**Contribution:** 2 fair

**Summary:**

This paper investigates the finite sum minimization problem where each function depends only on a subset of coordinates. The work combines cutting plane and interior point methods and achieves a better complexity in gradient oracles. As always, however, there is a catch.


**Questions:**

Some minor comments:

1. While we start with a problem (1.1) which is unbounded, we don't actually use it. The reformulation we use already assumes compactness. I think (1.1) only confuses. Also it uses $\theta$, while later we switch to $x$ variables, and we actually never show how this reformulation to (1.4) happens.

2. Table 1: In (1.1) we didn't assume strong convexity, while some methods in the table do. Thus, it is not clear what the authors want to convey. Also, what is the purpose of the last column? Subgradient methods handle this problem under non-smoothness.

3. line 50: It is not clear what "long chain of variable dependencies" is.

4. line 126-136: I suggest to be more instructive here. How exactly do you transform (1.1) into that form? What is the role of linear constraints $Ax=b$?

5. line 144-145: Inequalities (3.3) and (3.4) at this moment are not yet established, thus the explanation looks rather void.

6. Th. 2.3: We don't explain what is centroid here, however we provide a full line of text for the elementary definition of the halfspace.

7. line 192: No need to repeate notation of the polar set: it was defined just a few paragraphs above.



**Ethics Review Area:**

["I don’t know"]

**Limitations:**

-

**Strengths And Weaknesses:**

While the paper's contribution and motivation are definitely sound and interesting, especially from a mathematical point of view, I don't see its relevance to the NeurIPS community. For instance, the algorithm relies on the entropic barrier which is not implementable in general. Thus, the proposed complexity, in my opinion, is rather theoretical, with no hope to be practical.

This contribution also requires significant resources for review and hence for appreciation of the technique and with review workload we have this has been nearly impossible. Anyway, I think a venue with more focus on complexity would be a better fit for this work.

---

> ### Author Response · Authors · 2022-08-01
> **Response to Reviewer 5YUr**
>
> We thank the reviewer for their careful review and insightful comments. Below are our responses to specific comments/questions.
>
> ### Implementability and relevance to the NeurIPS community
>
> There has been a large body of work in the machine learning community that studies the complexity of convex optimization problems arising in practical machine learning problems. The convex formulation (1.1) is one of these and, in fact, has been studied widely in many previous papers in machine learning conferences under various other assumptions (e.g. smoothness, strong convexity, and other structural properties). Therefore, we believe a systematic study of this problem in the natural non-smooth setting is highly relevant to the NeurIPS community and could attract interest from theoreticians and practitioners alike.
>
> Historically, for many new settings in convex optimization, the first papers are often theoretical but they turn out to be important and lead to many follow-up works, some of which have significant practical impacts. For example, for the well-known fact that convex optimization can be solved in a polynomial number of first-order oracle calls, the famous center-of-gravity method takes exponential time. It took multiple decades of effort to get the runtime of such methods down to cubic time and these become the foundation of many recent exciting results in theoretical computer science. For example, acceleration techniques initially requires a complicated plane search in the 80s, but it is now used everywhere in machine learning. Also, the variance reduction trick initially requires full gradient memory but can now be done much more efficiently.
>
>
> In this paper, we introduced a new natural setting in convex optimization (1.1) and gave a very clean result in terms of its oracle complexity. While our paper is mostly theoretical, our result suggests the fast solvability of the problem (1.1) using first-order methods, and this is an important first step toward a more practical and implementable solution to (1.1). We believe like all previous theoretically novel results for new settings, there will be many follow-ups to our work and it will be of broad interest in the NeurIPS community.
>
> ### Our paper requires significant resources for review
>
> Due to the theoretical nature of our work, we tried to provide all the necessary context and preliminaries in Sections 1 and 2 including all relevant related work to better contextualize our result (e.g. Table 1). Our paper uses classical tools from convex analysis, interior-point methods, cutting-plane method, and convex geometry, and we believe this is actually a novelty of our techniques – similar strategies might also be useful for other non-smooth problems.
>
> ### Questions
> 1. (1&4) The purpose of the formulation (1.1) is to define a general problem with a sum structure. Boundedness is a standard (and often necessary) assumption for weakly polynomial convex optimization, e.g. cutting plane methods, and linear programming solvers that are based on interior-point methods. The switching from \theta to x is because in the transformation, the dimension of the problem increases (e.g. shared coordinates get copied, and we consider epigraphs).
>
> For the details of reformulation/reduction from (1.1) to (1.4), we currently have it in the proof of Thm B.11 (Ln759-770). The variables are changed from \theta to x. Since in this construction, we want to separate the same variable into different copies, the linear constraint $Ax=b$ is used to enforce that those variables are the same among all the copies. In our next version of the paper, we plan to explain this transformation more explicitly.
>
> 2. There are only a few prior techniques that can deal with non-smoothness (in particular, subgradient descent, cutting plane method and (robust) IPM). All these methods also immediately apply to more special settings (e.g. assuming smooth, strongly convex). The reason we include work with additional smoothness and strongly convex assumptions in Table 1 is to compare prior work in a broader context of the sum structure of (1.1), giving a sense of how the bounds for general non-smooth methods perform in comparison to methods designed under more structural assumptions.
> 3. For the long chain of variable dependencies, the example is given in eq. (1.3).
> 4. Responded together with 1.
> 5. We completely agree with this comment. We will give a more explicit explanation in our next version.
> 6. We will add the definition of centroid in our next version. The line is not for the purpose of define what is a halfspace, but is for the purpose of setting up the notations for the later statement of the theorem.
> 7. We completely agree and will just call Definition 1 (polar set) in our next version.
>
> We once again thank the reviewer for their time and effort and would be happy to better explain any part of our submission.

---

> > ### Comment · Reviewer_5YUr · 2022-08-07
> > **After rebuttal-1**
> >
> > While I am grateful to the authors for the historical excursion, I think the comparison with the center of gravity method, acceleration, or variance reduction is a pretty bold move. There are many things that can be done with the entropy barrier, you can't just say it's all relevant and interesting, but at the same time completely **unimplementable**.
> >
> >
> > I think it is clear from all reviews (including mine) that none of the reviewers has checked even 10% of the proofs. In my opinion, for majority of theoretical papers it is the technique that matters and not the final result.  I got the impression that the authors value their findings, so I'm even more surprised that they care more about publishing on NeurIPS than they do about fair review/evaluation.
> >
> >
> > And I want to *emphasize*, I am not against this paper. I can accept papers where I have not verified all proofs, but usually I understand why they work. Here I don't have even 10% of that confidence neither I was able to see it from other reviews.  Finally, I am sure it is possible to write a better paper from this submission which enlighten readers far better than it is now.

---

> > > ### Author Response · Authors · 2022-08-07
> > > **Response to After rebuttal-1by Reviewer 5YUr**
> > >
> > > We thank the reviewer for their additional comment. Below we address the major concerns in the reviewer's additional comment.
> > >
> > > ### Implementation Issue
> > >
> > > Note that the gradient and hessian of the entropy barrier are the centroid and covariance of some exponential distribution over a polytope.  In our setting, we only need a constant approximation of the centroid and covariance. In general, sampling from log-concave distribution takes n^4.2 time, but in our case, sampling from an exponential distribution over explicit polytope is much easier --- It takes around quadratic time in practice if the polytope is sparse, which holds for our case because of our assumption that each f_i only depends on a few variables.
> > >
> > > These results (and runtime complexity) of sampling from an exponential distribution over explicit polytope are not purely theoretical. For example, one problem in Bioinformatics about simulating metabolic networks (which contains tens of thousands of variables to hundreds of thousands of variables) involves computing centroid and covariance of exponential distribution over polytopes. As a result, they have been inventing new software for this task. One popular package for this problem is https://opencobra.github.io/cobratoolbox/stable/. It can sample from a polytope with tens of thousands of variables currently in seconds. In this package, there are ACHR, CHRR, RHMC, representing three iterations of improvement on the sampler in last 20 years. For example, for one human model (RECON3) with 12k variables, ACHR probably takes 10^8 cpu-sec to obtain one sample from that polytope while CHRR takes 10^4 cpu-sec and RHMC takes 30 sec.
> > >
> > > Therefore, we think it is evident that there could be an implementation of our algorithm which can perform reasonably well in practice.
> > >
> > > ### The authors value their findings
> > >
> > > We certainly highly value our results (which are almost optimal), but also our techniques. The techniques in our paper are not just an "application of entropic barrier". Our major technical contribution is a novel approach to combining interior point methods and cutting plane methods (largely developed independently in the optimization literature), achieving the best of both worlds in our setting.
> > >
> > > ### For the majority of theoretical papers, it is the technique that matters and not the final result
> > >
> > > While developing techniques in theory papers is certainly of fundamental importance, it is actually often the case that the results matter, especially in many of the first breakthroughs on big open problems. For instance, the first work by Spielman and Teng that Laplacian linear system can be solved in nearly linear time (aka Laplacian solvers) is very technical, not so readable, and definitely not implementable. But the result is of fundamental importance (it won the Godel prize), and has been the basis of later improvements on faster graph problem solvers (e.g. max-flow, matching, etc.). After decades of research on Laplacian solvers, we now know of much simpler and readable proof of Laplacian solvers (https://arxiv.org/abs/1605.02353) and packages (https://github.com/danspielman/Laplacians.jl).
> > >
> > > ### The authors care more about publishing on NeurIPS than they do about fair review/evaluation
> > >
> > > We do not agree with this comment (in fact, we think this comment is very subjective, and not relevant to actually providing a fair review/evaluation of our paper). The authors care a lot about fair review/evaluations, and that is why we provide detailed responses to every comment of all reviewers.

---

> > > > ### Comment · Reviewer_5YUr · 2022-08-07
> > > > **Thanks for the response**
> > > >
> > > > With all respect to the Spielman and Teng work, I would reject that paper from NeurIPS because of similar reasons. There are different venues for different papers, that's why they have chosen a tcs conference/journal and not ml conference.
> > > >
> > > > I do appreciate the obtained result. But I want to appreciate the technique as well and want to be at least half-sure that it is correct or see such a level of confidence from other reviewers.
> > > >
> > > > > The authors care a lot about fair review/evaluations, and that is why we provide detailed responses to every comment of all reviewers
> > > >
> > > > Let's be realistic. There isn't a lot of difficult comments here :) By fair review I meant the one where it is clear for authors that reviewers did check some parts of the proofs.
> > > >
> > > > But please don't take me so negatively. I am just more concerned about the impact of the article than its publication status.

---

### Official Review · Reviewer_wH1T · 2022-07-12

**Rating:** 7
**Confidence:** 3
**Soundness:** 4 excellent
**Presentation:** 4 excellent
**Contribution:** 4 excellent

**Summary:**

This work developed a new algorithm with near linear gradient oracle complexity (in terms of total effective dimension) for finite-sum non-smooth convex optimization, which appears to be the first work of achieving this goal.

**Questions:**

See strengths and weaknesses.

**Limitations:**

The authors have adequately addressed the limitations of their work.

**Strengths And Weaknesses:**

Strengths:

- The proposed algorithm is based on a novel combination of cutting-plane methods and interior-point methods, which is technically non-trivial. The method achieves near linear gradient oracle complexity in terms of total effective dimension, which matches the theoretical lower bound. The result drastically improves upon the complexity of directly applying cutting-plane method when the gradients are sparse. Overall the paper is well written.

Weaknesses:

- It would be great if the authors can provide a discussion on prior works that deal with non-smooth empirical risk minimization problem (i.e., ERM with a non-smooth regularizer that admits an efficiently computable proximal operator), which helps understanding the positioning of this work.

---

> ### Author Response · Authors · 2022-08-01
> **Response to Reviewer wH1T**
>
> We thank the reviewer for their careful review of our paper and for providing insightful comments. Below is our response to their specific question.
>
> The work on non-smooth ERM crucially requires the objective function to be a sum of a smooth ERM part and a non-smooth regularizer. There are many important problems for which the objective function cannot be split in this way (such as the examples provided in our paper), so for these problems, the techniques developed for non-smooth ERM don’t apply. In comparison, our work can be understood as dealing with a more general ERM problem with fewer structural assumptions.
>
> We thank the reviewer for pointing this out and will add this discussion to the related work section in our revised manuscript.

---

### Official Review · Reviewer_Q9he · 2022-08-01

**Rating:** 6
**Confidence:** 4
**Soundness:** 3 good
**Presentation:** 3 good
**Contribution:** 3 good

**Summary:**

This paper studies a class of decomposable convex optimization problem. An algorithm with optimal query complexity is proposed. In addition, the authors also improve the evaluation oracle complexity for decomposable submodular minimization.

**Questions:**

What is the definition of $\mathcal{K}_i$ that appeared in Section 1.2 and 1.3?

**Limitations:**

This is a theory paper and I believe there is no potential negative societal impact.

**Strengths And Weaknesses:**

Strengths: This paper makes solid contribution to decomposable convex optimization problem with possibly non-smooth objective. The authors propose new algorithm for this problem with optimal query complexity. The proof technique in this paper is novel.

Weaknesses: The organization and presentation of this paper needs to be improved.
1. I believe the main algorithm should be presented more clearly and explicitly in the main text, namely one should be able to implement Algorithm 1 after reading Section 3. Currently Section 3.1 seems like a mixture of implementing details and algorithmic intuitions, but due to space limitation I think the authors are not able to present either of them clearly. For example, how to set the initialization K_in, K_out and x_initial? The readers are referred to Appendix C.1 and C.2, where they might only find theorems and proofs instead of implementing details. What is the expression of A, b and c using the quantities already introduced in the main text? Can each step in Algorithm 1 be computed efficiently?
2. I think at least some part of Section 2 is not necessary for the main text, which can be moved to the appendix and save space for presenting and explaining algorithmic details.
3. I don't see a reason why the authors only present informal theorem (Theorem 1.2) in the main text and defer the formal theorem to the appendix. It seems to me that the formal theorem is not too complicated.

---------------After rebuttal----------------------

I would like to thank the authors for their response. Given that all my comments and authors' response leads to revision in the next version of the paper, I decide to keep my score and leaning to an accept.

---

> ### Author Response · Authors · 2022-08-01
> **Response to Reviewer Q9he**
>
> We thank the reviewer for their careful review of our paper and for providing insightful comments. We completely agree with all the comments. Below are our responses to specific comments/questions.
>
> 1. Indeed the current algorithm implementation details and intuitions are mixed and could be confusing. In our next version of the paper, we plan to improve this part of the presentation by breaking this section into two parts: in the first part we only give the algorithm implementation details and pseudocode, with all notations like K_in, K_out and x_initial, A, b, c explicitly defined and more details on how the algorithm could efficiently make a step; while in the second part, we plan to give some algorithmic intuitions (and in particular, shed light on the high-level analysis in Section 4).
>
> 2. This is a great idea to save some space for other sections. In our next version, we plan to put all the preliminary lemmas that are only invoked in the appendix to a Preliminaries section in the appendix
>
> 3. Indeed, we will present the formal theorem in Theorem 1.2 in our next version.
>
> ### Questions
>
> K_i is the epigraph of the function f_i. This is currently only mentioned until the beginning of Section 3, and we plan to add this definition to Section 1.2 before the first occurrence of K_i.
>
> We again thank the reviewer for their time and effort in reading the paper and for providing very insightful comments.

---

### Meta-Review · Area_Chair_E4jG · 2022-08-20

**Recommendation:** Accept
**Confidence:** Certain

**Metareview:**

This paper studies the finite-sum optimization problem with convex yet possibly non-smooth objective functions. The authors develop an efficient algorithm that allows one to solve this problem with a nearly linear number of calls to the gradient oracle (in terms of the effective dimensionality), thus improving upon prior art.  The key idea lies in combining the cutting-plane method and the interior-point method in a novel manner. While a reviewer has raised concerns about implementation of the proposed algorithms, the theoretical contributions of this paper are solid and important, and hence I recommend acceptance. It would still be great if the authors could consider discussing practical implementation of the proposed algorithms.

**Award:**

No

---

### Decision · Program_Chairs · 2022-09-14

Accept